# miR-483-5p offsets functional and behavioural effects of stress in male mice through synapse-targeted repression of *Pgap2* in the basolateral amygdala

Mariusz Mucha [1,9] ✉, Anna E. Skrzypiec[1,9], Jaison B. Kolenchery[1,9], Valentina Brambilla[1], Satyam Patel[2], Alberto Labrador-Ramos[1], Lucja Kudla[3], Kathryn Murrall[1], Nathan Skene [4,5], Violetta Dymicka-Piekarska[6], Agata Klejman[7], Ryszard Przewlocki[3], Valentina Mosienko [1,8] ✉ & Robert Pawlak[1]

Severe psychological trauma triggers genetic, biochemical and morphological changes in amygdala neurons, which underpin the development of stress-induced behavioural abnormalities, such as high levels of anxiety. miRNAs are small, non-coding RNA fragments that orchestrate complex neuronal responses by simultaneous transcriptional/translational repression of multiple target genes. Here we show that miR-483-5p in the amygdala of male mice counterbalances the structural, functional and behavioural consequences of stress to promote a reduction in anxiety-like behaviour. Upon stress, miR-483-5p is upregulated in the synaptic compartment of amygdala neurons and directly represses three stress-associated genes: *Pgap2*, *Gpx3* and *Macf1*. Upregulation of miR-483-5p leads to selective contraction of distal parts of the dendritic arbour and conversion of immature filopodia into mature, mushroom-like dendritic spines. Consistent with its role in reducing the stress response, upregulation of miR-483-5p in the basolateral amygdala produces a reduction in anxiety-like behaviour. Stress-induced neuromorphological and behavioural effects of miR-483-5p can be recapitulated by shRNA mediated suppression of *Pgap2* and prevented by simultaneous overexpression of miR-483-5p-resistant *Pgap2*. Our results demonstrate that miR-483-5p is sufficient to confer a reduction in anxiety-like behaviour and point to miR-483-5p-mediated repression of *Pgap2* as a critical cellular event offsetting the functional and behavioural consequences of psychological stress.

Severe or prolonged stress often leads to the disruption of neuronal homoeostasis, triggering the development of physiological and behavioural abnormalities such as elevated levels of anxiety[1–4]. Anxiety disorders (encompassing generalised anxiety disorder, panic attacks, phobias, obsessive-compulsive disorder and post-traumatic stress disorder) are the most common psychiatric conditions currently diagnosed, affecting ~25% of the population at least once in their lifetime[5,6]. The enormous economic and social impact of anxiety

disorders can be attributed to the low efficiency of available anxiolytic therapies, with up to 50% of patients not achieving full remission despite multiple medication attempts[6]. Limited success in developing effective anxiolytic drugs stems from insufficient knowledge of the neural circuits of anxiety and molecular events underpinning stress-related neuropsychiatric states.

Various lines of evidence from studies in rodents indicate that the brain region critical for the development of stress-induced anxiety is the amygdala The amygdala collects spatial and sensory information about the surrounding environment through hippocampal, thalamic and cortical inputs arriving at its basolateral division (BLA)[7–9]. Stress-related neuronal computations are subsequently conveyed to the central amygdala (CeA) and the bed nucleus of stria terminalis (BNST)[7,8]. CeA and BNST relay emotionally relevant neural signals to the downstream brain centres responsible for the expression of behavioural reactions (e.g. risk avoidance, low food intake, social avoidance, increase in blood pressure and heart rate) that collectively assemble anxiety-like states[7,8,10].

At the cellular level, the amygdala is composed of a variety of neuronal subtypes with diverse genetic and connectivity profiles. Broadly speaking, amygdala neurons can be classified into excitatory glutamatergic or inhibitory GABA-ergic neurons, which can be further divided into smaller populations that use different neuromodulators[7,11]. Consequently, full comprehension of the amygdala's function requires an understanding of the molecular machineries operating within amygdala sub-divisions, and their contributions towards specific behavioural outcomes.

One of the molecular factors modulating neuronal functioning is a family of small, non-coding RNA sequences called microRNAs (miRNAs)[12]. miRNAs destabilise messenger RNAs (mRNAs) or repress translation of target molecules, primarily through binding to the 3'UTR of complementary genes[13]. Due to their unique capacity to simultaneously alter the expression of numerous transcripts, miRNAs can act as hub regulators of cellular functions at the pathway level. The ability of miRNAs to spatiotemporally regulate gene expression makes them strategically poised to control complex neuropsychiatric conditions, such as pathological anxiety[12,13]. The role of amygdala miRNAs in these phenomena is not well-understood. miRNAs contribute to the development of stress-related neuropsychiatric states, such as anxiety, fear and depression[13–19], but the molecular and cellular mechanisms they utilise to regulate stress resilience and susceptibility are largely unknown. Our study aimed to identify a specific subpopulation of the mouse amygdala miRNAs activated selectively by acute psychological stress and to decode the molecular pathways they participate in to regulate the stress response.

Here we found that miR-483-5p is a critical regulator of stress-related dendritic arbour and dendritic spine plasticity in the amygdala and attenuates the development of anxiety-like behaviour. Upon stress, miR-483-5p is upregulated in the synaptic compartment of amygdala neurons and represses three stress-associated genes (*Pgap2*, *Gpx3* and *Macf1*) through binding to 3'UTRs of their respective mRNAs. Upregulation of miR-483-5p leads to selective shrinkage of distal dendrites and promotes the formation of mature, mushroom-like dendritic spines crucial for emotional memory formation, processing and storage. Consistent with its pivotal role in preventing functional and behavioural consequences of stress, the upregulation of miR-483-5p in the basolateral amygdala is sufficient to produce an anxiolytic effect. This effect can be mimicked by either lentiviral overexpression of the miR-483-5p or by shRNA-mediated suppression of the *Pgap2* gene. Overexpression of both miR-483-5p and miR-483-5p-resistant *Pgap2* prevents the effects of miR-483-5p on stress-induced changes to spine morphology and behaviour.

Our results identify the miR-483-5p-mediated repression of *Pgap2* as a critical molecular event reducing the functional and behavioural consequences of psychological stress.

## Results

### miR-483-5p is upregulated by stress in the amygdala and enriched in the synaptic compartment

Psychological stress can cause profound alterations in gene expression profiles in several brain regions, including the amygdala[20]. One mechanism by which such prominent changes can be achieved is through the regulation of miRNA levels, which simultaneously regulate numerous target genes[12]. In order to investigate how stress affects miRNA expression in the amygdala, we subjected adult (8–12-week old), male C57BL/6J mice to acute 6-h restraint stress. Immediately after this we dissected their amygdalae, extracted the RNA and performed microarray analysis (Fig. 1a–c). We found five miRNAs (miR-1192, miR-1224, miR-1892, miR-1894-3p and miR-483-5p) to be significantly upregulated, with miR-483-5p reaching the highest levels (~2.5-fold increase; $p < 0.001$; Fig. 1d, e). Quantitative real-time PCR confirmed the upregulation of all five miRNAs, including miR-483-5p, by at least 2-fold in the amygdala upon stress (Fig. 1f; $p = 0.0006$ for miR-483-5p). Based on the highest level of upregulation our subsequent studies focused on the role of amygdalar miR-483-5p in the stress response and on its mechanism of action.

To investigate whether the expression of miR-483-5p is within the neuronal component we performed fluorescence in situ hybridisation (FISH) assay targeting miR-483-5p molecules followed by immuno-histochemistry for neuronal marker NeuN in the amygdala. We found that following restraint stress, the miR-483-5p is expressed predominantly within the neuronal component of the amygdala (Fig. 2a, b). miRNAs can either act in the diffused cytosol or locally in discrete cellular compartments, such as the synaptic region, where they may precisely regulate protein synthesis in an activity-dependent manner[21,22]. Such spatially restricted regulation of protein expression allows subtle control of functional and structural features of synapses in an input-specific fashion. To investigate whether miR-483-5p is present in the synaptic compartment upon stress, we performed a FISH assay followed by immunostaining for synaptic marker HOMER using restraint-stressed animals (Fig. 2a). We found that miR-483-5p is present in the subset of synapses as demonstrated by its co-localisation with HOMER.

To investigate whether stress-induced upregulation of miR-483-5p is subject to compartmental bias, we quantified the expression of miR-483-5p in the cytosolic and synaptosomal fractions isolated from amygdalae of mice subjected to restraint stress and stress-naïve controls. First, we found a 4-fold enrichment of miR-483-5p in the synaptosomal fraction extracted from stress-naïve mice compared to their cytosol counterparts (Fig. 3a). We also observed that stress caused a further 4-fold increase of miR-483-5p specifically in the synaptosomes (Fig. 3a; $p = 0.0079$ stress vs control in synaptosome fraction). These results indicate that miR-483-5p is enriched in the synaptic compartment, where it is locally regulated by stress, thereby providing a possible mechanism for spatially restricted control of protein synthesis that may affect neuronal arborisation and dendritic spine morphology.

### miR-483-5p promotes dendritic shrinkage and the formation of mushroom-like spines in amygdala neurons

Local enrichment and stress-induced upregulation of miR-483-5p in synaptosomes prompted us to investigate if it plays a role in the regulation of structural plasticity associated with this neuronal compartment. It has been previously demonstrated that stress causes region-dependent changes in dendritic arborisation, manifested as shrinkage of apical dendrites in the CA3 region of the hippocampus and dendritic outgrowth of the basolateral amygdala neuronal dendritic arbour[4,23,24]. Because the role of miR-483-5p in these phenomena was unknown we next studied the effects of miR-483-5p on the morphology of amygdala neurons. Primary mouse amygdala neurons were double-transfected with a vector containing miR-483-5p (a scrambled sequence was used as control) along with the GFP reporter gene, and a plasmid driving the

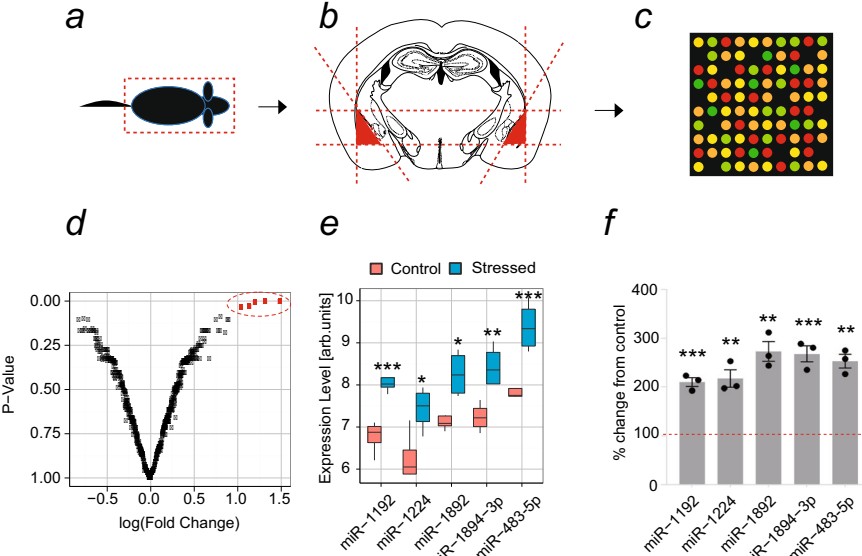

**Fig. 1 | Regulation of miRNAs in mice by restraint stress in the basolateral amygdala. a** C57BL/6J mice were subjected to a 6-h of restraint stress or control, stress-naïve animals were left undisturbed in their home cages. Mice were then anaesthetised and (**b**) miRNA was isolated from dissected basolateral amygdalae (**c**). The resulting miRNA was then hybridised onto microarrays to examine miRNAs' expression levels. **d** Bioinformatics analysis of the microarray data revealed upregulation of miR-1192, miR-1224, miR-1892, miR-1894-3p and miR-483-5p in the basolateral amygdala in mice after stress as highlighted by red dotted line in the volcano plot. **e** Pre- (red rectangles) and post-stress (blue rectangles) expression levels identified miR-483-5p as the most significantly elevated miRNA. Data are presented on a box and Whiskers plot where the box extends from 25th to 75th percentile, and whiskers range from minimum and maximum value, with the centre denoting the median value; $n = 4$ microarrays where each array was hybridised with pulled mRNA of 3 individual animals (12 animals in total); statistical significance is assessed by two-sided unpaired $t$ test, $*p < 0.05$, $**p < 0.01$, $***p < 0.001$ vs control mice (**f**) To verify the microarray results, miRNA isolated from basolateral amygdalae of mice subjected to 6-h immobilisation was analysed by quantitative RT-PCR using miRNA-specific probes. qRT-PCR confirmed that miR-483-5p was among the miRNAs most prominently induced by stress in the basolateral amygdala in mice. Data are presented as dot plots with mean ± SEM; $n = 3$ animals per group; statistical significance is assessed by two-sided unpaired $t$ test, $**p < 0.01$, $***p < 0.001$ vs control mice. Source data and the exact $p$-values are provided in a Source Data file.

expression of the fluorescent protein tdTomato to highlight dendritic branches. The neurons were visualised by confocal imaging and the complexity of the dendritic arbour was examined (Fig. 3b, c). Sholl analysis revealed that miR-483-5p caused selective shrinkage of the distal dendritic segments (130–160 μm from the soma) by 50–70% ($p$ (130 μm) = 0.045, $p$ (140 μm) = 0.0185, $p$ (150 μm) = 0.0042, $p$ (160 μm) = 0.0026), while the arborisation of the proximal dendrites (10–120 μm from the soma) was indistinguishable from the control group (Fig. 3b, c).

Rearrangements of neuronal tree complexity can coincide with further functional alterations of the neuronal activity either by a change in density or morphology of dendritic spines/synapses. To investigate this possibility, we examined whether miR-483-5p triggers changes in the dendritic spine morphology, as suggested by its expression within the synaptic compartment. In order to address this, we co-transfected mouse primary amygdala neurons with plasmids expressing miR-483-5p-pIRES-mCherry and AcGFP1-GFP (a membrane-targeted GFP consisting of N-terminal 20 amino acids of neuromodulin fused with AcGFP1[25]) to visualise dendritic spines. The analysis performed accordingly to previously published criteria[20,25], revealed a ~60% increase in the proportion of synapse-prone mature mushroom spines (Fig. 3d, e; $p < 0.0001$) accompanied by a ~60% decrease in the percentage of neuroplastic filopodia-like structures (Fig. 3d, e; $p < 0.0001$). No significant differences in thin, stubby or branched spines were observed after the overexpression of miR-483-5p.

To verify whether similar dynamic rearrangements of dendritic spine subclasses take place in situ, we stereotaxically injected lentiviral particles expressing miR-483-5p-EGFP (or control, scrambled sequence co-expressed with EGFP) into mouse amygdalae. We found that upon viral injection followed by restraint stress, the level of miR-483-5p expression was consistently 4-fold higher than in the amygdalae of animals injected with the control virus (Supplementary Fig. 5a).

Taking advantage of the co-expressed EGFP to identify the transduced neurons we placed microcrystals of DiI (a lipophilic cationic indocarbocyanine dye) onto the neuronal cell bodies to allow visualisation of their dendritic spine formations. We found that expression of miR-483-5p triggered a shift in the dendritic spine subclasses' composition, towards mature, mushroom-like forms at the expense of stubby spines (Fig. 3g). We found no significant changes in dendritic spine density in cultured neurons (Fig. 3f) while a small increase in spine density was found in situ neurons overexpressing miR-483-5p (Fig. 3g, h) indicating that miR-483-5p on its own promotes maturation of mushroom-like, synapse-prone spines while increasing spine density following psychological stress.

### miR-483-5p represses a panel of stress-related amygdala target genes

miRNAs can affect a variety of neuronal functions defined by the roles of their target genes[13,14,16,19,22]. In order to identify the genes regulated by a miR-483-5p expression, we undertook a bioinformatics/data mining approach. We have selected common amygdala-expressed transcripts listed in the ElMMo database (215 hits identified), stress-related targets identified by the AmiGo search engine (4816 hits) and miR-483-5p targets found through the TargetScan algorithm (525 hits) (Fig. 4a). Cross-comparison identified 12 genes present in all three databases (Fig. 4a), which we subsequently selected for experimental analysis (Fig. 5a). All identified genes contain in their 3'UTRs evolutionary conserved miR-483-5p target sequence (Fig. 6c, d).

In order to examine which of the 12 putative miR-483-5p targets were genuine, we overexpressed miR-483-5p (or a scrambled control sequence) in Neuro2a cells and measured the expression of the target genes' mRNAs by qRT-PCR. We found that under basal conditions miR-483-5p significantly suppressed the expression of only one out of twelve putative targets (*Gpx3*) (Fig. 4b, $p = 0.0013$; Supplementary

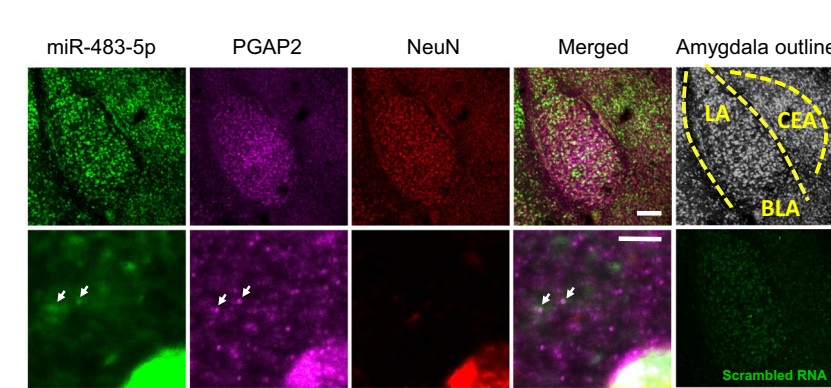

*a*

**Fig. 2 | miR-483-5p is expressed in amygdala neurons in mice upon restraint stress.** Fluorescence in situ hybridisation (FISH) targeting miR-483-5p (green) and immunohistochemistry for neuronal marker NeuN (**a**, red) and synaptic marker HOMER (**a**, purple) or PGAP2 (**b**, purple) revealed miR-483-5p expression on amygdala neurons following restraint stress. The arrows indicate the colocalization of the miR-483-5p with HOMER (**a**) or PGAP2 (**b**). Scale bar = 200 μm (upper panels) and 5 μm (lower panels). Representative images, experiments were performed independently with similar results on 3 animals. LA−lateral amygdala, BLA−basolateral amygdala, CEA−central amygdala.

Fig. 1). We reasoned that, since miR-483-5p is induced by stress it may require the presence of stress hormone(s) in order to exert its full effect. To investigate this possibility, we treated miR-483-5p-transfected Neuro2a cells (or cells transfected with scrambled sequence as controls) with a corticosterone analogue dexamethasone before analysing the target gene expression. We found that dexamethasone stimulated the expression of eight out of twelve putative target genes, and miR-483-5p suppressed the expression of three of these (*Pgap2, Gpx3 and Macf1*) (Fig. 4b and Supplementary Fig. 1). Overall, our results demonstrate that miR-483-5p controls the expression of three stress-related genes in a stress hormone-dependent manner.

Next, we postulated that if synaptic miR-483-5p controls the stress response through regulation of *Pgap2, Gpx3 and Macf1* expression, then these transcripts should be downregulated upon stress in the matching cellular compartment of amygdala neurons. To verify that we subjected mice to restraint stress and analysed the levels of coding mRNA extracted from the whole cellular lysate or synaptosomal fractions (Fig. 5a, b). qRT-PCR revealed a sharp 50−60% decrease in the expression of two genes in the cellular lysate (*Pgap2* and *Macf1*, Fig. 5b), while only mRNA coding for *Pgap2* was significantly downregulated in the synaptosomal compartment (Fig. 5b; $p = 0.0119$).

Synaptic enrichment and regulation of dendritic spine morphology by miR-483-5p prompted us to investigate which of the proteins encoded by miR-483-5p target mRNA exhibit a membrane and/or

synaptic expression pattern, prerequisite for synapse-specific effects of miR-483-5p in the amygdala.

Immunohistochemistry confirmed the prominent presence of all three target proteins in the basolateral amygdala (Fig. 5c). Multi-label immunostaining using antibodies against the target proteins, along with the neuronal marker NeuN and astrocyte marker glial fibrillary acidic protein (GFAP), revealed exclusive neuronal expression of *Pgap*2 and *Macf*1, while *Gpx*3 was present in both neurons and astrocytes in amygdala. All identified target proteins were present in the cellular membrane in agreement with their presumptive synaptic function. Additionally, *Pgap*2 also showed a diffuse, puncta signal outside of the neuronal bodies possibly within synaptic formations (Figs. 2b, 5c).

To additionally verify the interaction between miR-483-5p and its putative target genes we have performed an in vitro luciferase assay[16] where the 3′ UTR of *Pgap2*, *Gpx3* and *Macf1* mRNAs were subcloned into the pmiRGLO Dual-Luciferase miRNA-target expression vector, downstream of the *Firefly* luciferase reporter gene. Upon co-transfection into Neuro2a cells the 3′UTR constructs with the miR-483-5p-expressing (or scrambled sequence control) plasmids, the intensity of luminescence produced by *Firefly* luciferase was measured and normalised to independently expressed *Renilla* luciferase. All examined constructs demonstrated a significant reduction in the firefly luciferase activity upon miR-483-5p expression, suggesting that *Pgap2*, *Gpx3* and *Macf1* were suppressed by miR-483-5p binding (Fig. 6b; $p(Pgap2) = 0.0243$, $p(Gpx3) = 0.0424$, $p(Macf1) = 0.0097$).

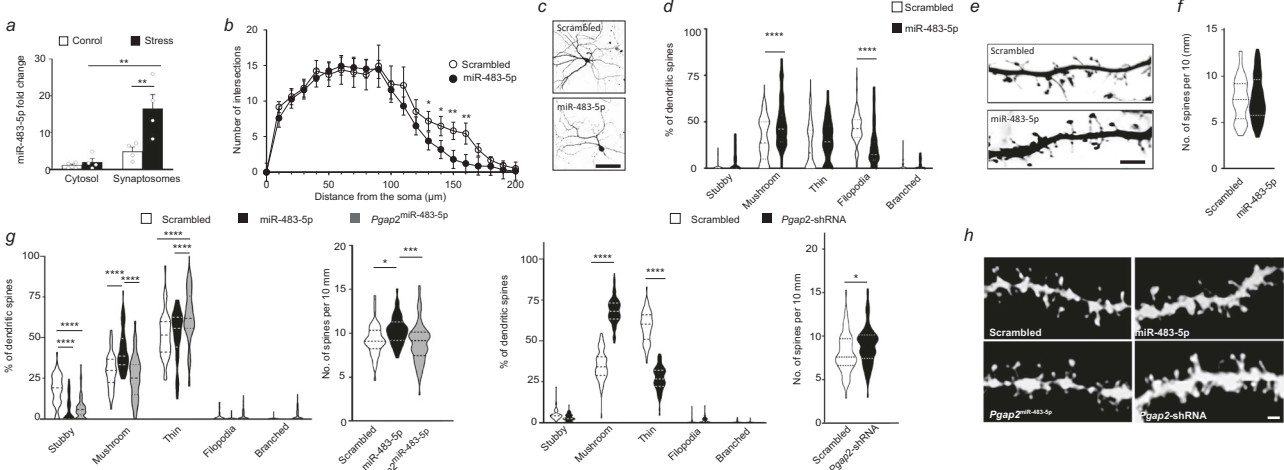

**Fig. 3 | miR-483-5p is induced by stress in mice in the juxtasynaptic compartment to control dendritic arborisation and spine morphology. a** Mice (stress-naïve or subjected to 6-h immobilisation stress) were anaesthetised, their baso-lateral amygdalae were dissected and the cytosol and synaptosomal fractions were isolated. miRNAs were extracted from both fractions, reverse transcribed, and their expression levels analysed by quantitative RT-PCR. The study revealed that stress elevated the expression of miR-483-5p predominantly in the synaptosomal fraction. Data are presented as dot plots with mean ± SEM; $n = 4$ animals per group; statistical significance is assessed by 2-way ANOVA with Tukey's multiple comparisons test as a post-hoc test, $**p < 0.01$ vs an indicated group. **b** Sholl analysis of neuronal structure revealed that overexpression of miR-483-5p (mimicking its stress-induced upregulation) in primary mouse amygdala neurons, caused shrinkage of the distal parts of the dendritic arbour (130–160 μm from the soma). Data are presented as mean ± SEM, $n_{neurons}$ (scrambled) = 21; $n_{neurons}$ (miR-483-5p) = 34 neurons; statistical significance is assessed by mixed effect model with 2-step linear procedure of Benjamin, Krieger and Yekuteli for individual comparisons, $**p < 0.01$, $*p < 0.05$ vs scrambled. **c** Representative images of primary amygdala neurons transfected with either the miR-483-5p or scrambled control vector. Scale bar = 50 μm. **d** Overexpression of miR-483-5p in primary mouse amygdala neurons caused an increase in the proportion of mature, mushroom-like spines, with a simultaneous decrease in the immature, filopodia-like protrusions. Data are presented on a violin plot where the centre denotes median value; $n_{neurons}$ (scrambled) = 20 and $N_{regions}$ (scrambled) = 59; $n_{neurons}$ (miR-483-5p) = 21 neurons and $N_{regions}$ (miR-483-5p) = 54, statistical significance is assessed by 2-way ANOVA with

Bonferonni's correction as a post-hoc test, $****p < 0.0001$ vs scrambled. **e** Representative images of dendritic spines in primary mouse amygdala neurons transfected with either the miR-483-5p or scrambled control vector. Scale bar = 5 μm. **f** The density of dendritic spines in primary mouse amygdala neurons was not affected by the overexpression of miR-483-5p. Data are presented on a violin plot where the centre denotes median value, statistical significance is assessed by two-sided unpaired $t$ test, $n_{neurons}$ (scrambled) = 20 and $N_{regions}$ (scrambled) = 59; $n_{neurons}$ (miR-483-5p) = 21 neurons and $N_{regions}$ (miR-483-5p) = 54. **g** Mice were first injected with lentiviruses carrying miR-483-5p-EGFP, $Pgap2$-shRNA-EGFP or $Pgap2^{miR-483-5p}$-EGFP ($Pgap2$ resistant to miR-483-5p) into basolateral amygdala, and following 6-h immobilisation dendritic spines were visualised using DiI staining. Expression of both miR-483-5p and $Pgap2$-shRNA resulted in increased proportion of mature mushroom-shaped spines, while no change to a population of mushroom spines were found following expression of miR-483-5p-resistant $Pgap2$ when compared to the control. $N = 3$ animals per group, 1.6–5.5 mm of the dendrites per group, $n = 338$–4767 spines per group, $n_{regions}$ (scrambled) = 50, $n_{regions}$ (miR-483-5p) =45, $n_{regions}$ ($Pgap2$) = 78. Expression of both miR-483-5p and $Pgap2$-shRNA lead to about 10% increase of dendritic spine density. Data are presented on a violin plot where the centre denotes median value; statistical significance is assessed by 2- (graphs 1 and 3) or 1-way ANOVA (graph 2) or by two-sided $t$ test (graph 4), $****p < 0.0001$, $***p < 0.001$, $**p < 0.01$, $*p < 0.05$ vs an indicated group. **h** Representative images of DiI stained neurons. Scale bar = 2 μm. Source data and the exact $p$-values are provided in a Source Data file.

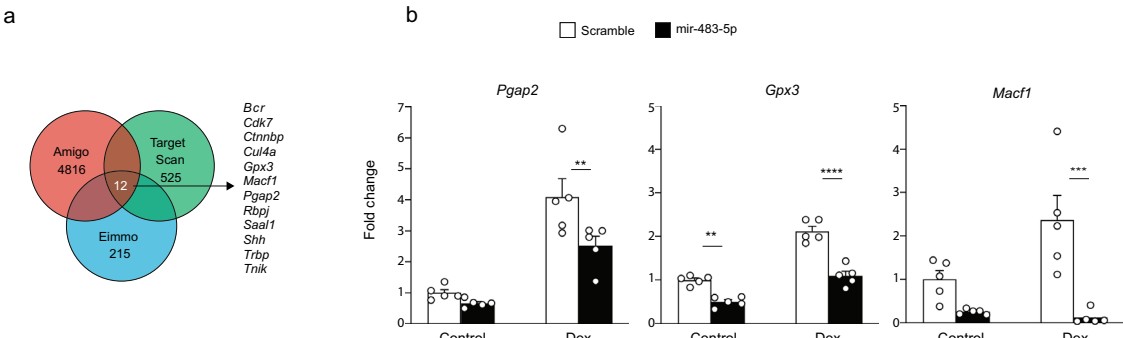

**Fig. 4 | Identification and characterisation of stress-related genes in the amygdala. a** Bioinformatics database cross-comparison identified 12 genes expressed in the amygdala (the EIMMo database), which are also regulated in response to stress (the AmiGo search engine) as potential targets of miR-483-5p (the TargetScan algorithm). **b** To experimentally identify genuine stress-related miR-483-5p targets, Neuro2a cells were transfected with the miR-483-5p-expressing vector (or scrambled control vector) and gene expression levels following

dexamethasone were measured by quantitative RT-PCR (see also Supplementary Fig. 1). The expression of $Pgap2$, $Gpx3$ and $Macf1$ mRNAs was suppressed by miR-483-5p, confirming these genes as genuine miR-483-5p targets. Data are presented as dot plots with mean ± SEM; $n = 5$ animals per group; statistical significance is assessed by 2-way ANOVA with Šídák's multiple comparisons test as a post-hoc test, $****p < 0.0001$, $***p < 0.001$, $**p < 0.01$ vs an indicated group. Source data and the exact p-values are provided in a Source Data file.

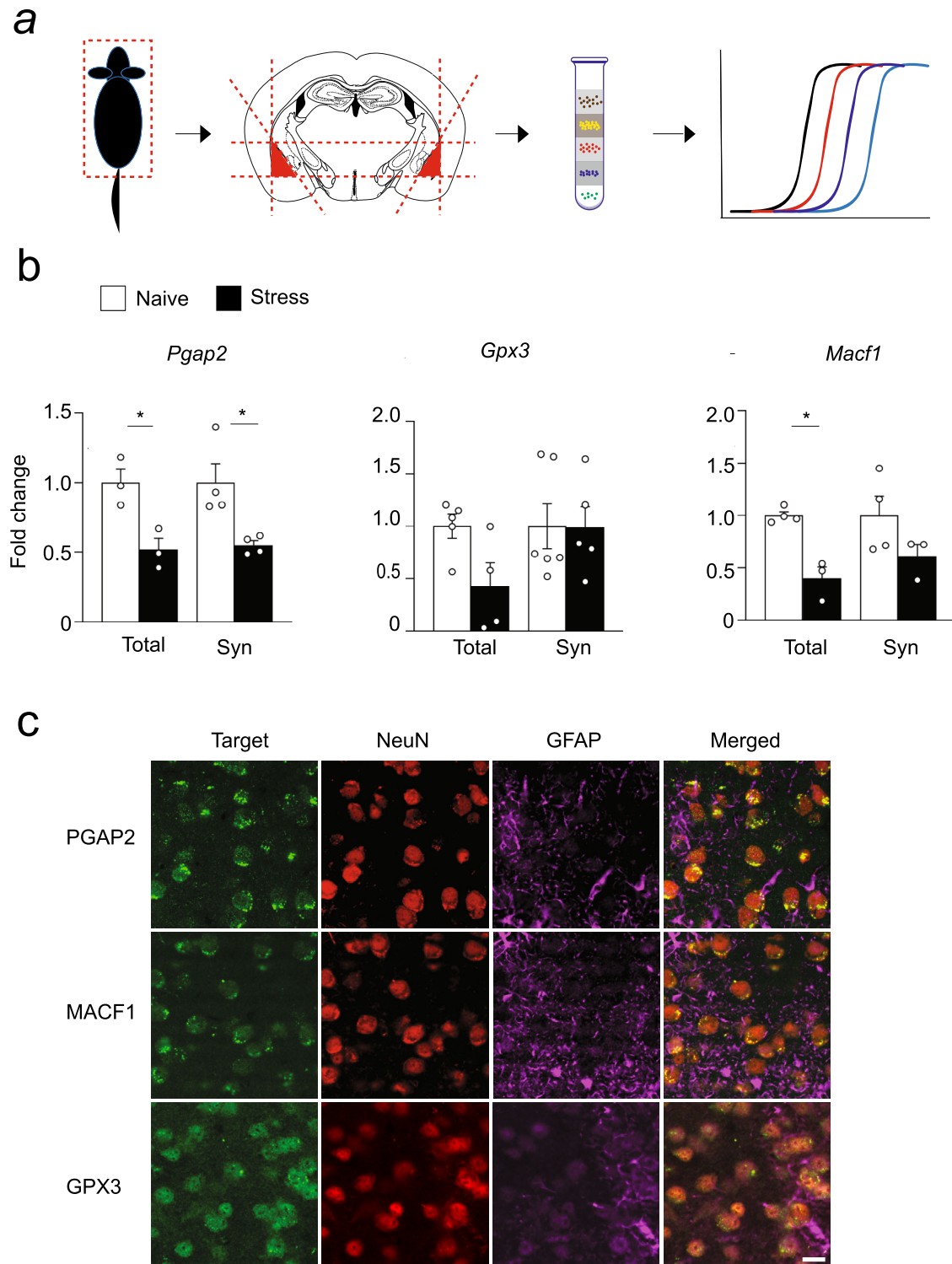

**Fig. 5 | miR-483-5p controls set of target gene expression in the juxtasynaptic compartment of amygdala neurons. a** C57/BL/6J mice were subjected to 6-h immobilisation (the stress-naïve group remained in their home cages), anaesthetised, and their basolateral amygdalae dissected rapidly on ice from coronal brain sections (dissection boundaries shown as red dotted lines) and the cytosol and synaptosomal fractions were isolated. mRNAs were extracted from both fractions, reverse transcribed, and *Pgap2*, *Gpx3* and *Macf1* gene expression was measured by quantitative RT-PCR. **b** qRT-PCR revealed 50–60% stress-induced decrease in the expression of *Pgap2* and *Macf1* in the whole-cell amygdala homogenate. In contrast only the expression of *Pgap2* was suppressed in the synaptosomal fraction. Data are presented as dot plots with mean ± SEM; $n = 3$–6 animals per group; statistical significance is assessed by 2-way ANOVA with Šídák's multiple comparisons test as a post-hoc test, *$p < 0.05$ vs an indicated group. Source data and the exact *p*-values are provided in a Source Data file. **c** Immunohistochemistry revealed that PGAP2, MACF1 and GPX3 proteins are present in the neurons in the direct vicinity of the cell membrane. Representative images, experiments were performed independently with similar results on 3 animals. Scale bar = 20 μm.

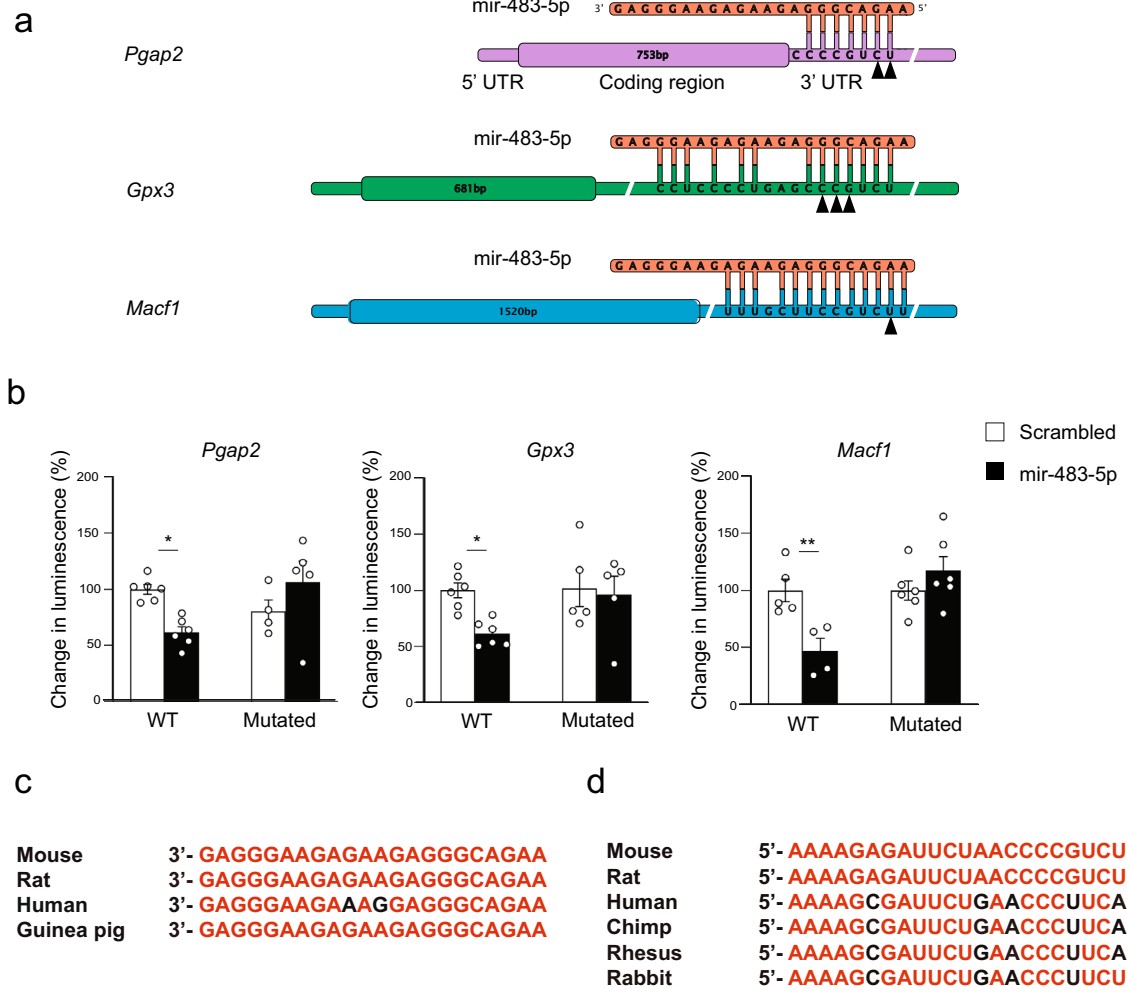

**Fig. 6 | miR-483-5p controls the expression of *Pgap2*, *Gpx3* and *Macf1* genes directly through binding to 3' UTR regions. a** Alignment of miR-483-5p sequence (red) with 3'UTRs of mouse *Pgap2*, *Gpx3* and *Macf1* mRNA sequences (violet, green and blue, respectively). Complementary nucleotides within miR-483-5p and the 3'UTRs of its target mRNAs are connected with vertical lines. Arrowheads point to the nucleotides within the 3'UTR seed match sequences mutated in order to disrupt miR-483-5p binding. **b** To perform in vitro luciferase assay the 3'UTR of *Pgap2*, *Gpx3* and *Macf1* mRNA were inserted into the pmiRGLO Dual-Luciferase miRNA-target expression vector, downstream of the firefly luciferase reporter gene. Neuro2a cells were double-transfected with the above vector together with the miR-483-5p-expressing (or scrambled control) vector, and the intensity of the luminescence produced by firefly luciferase was measured. The signal was normalised to the

Renilla luciferase luminescence. In all three cases miR-483-5p expression reduced the Firefly luciferase activity, confirming that miR-483-5p suppressed *Pgap2*, *Gpx3* and *Macf1*. Mutating the seed match sequences of the 3'UTR (arrowheads in a) abolished miR-483-5p-mediated suppression, confirming that miR-483-5p affects expression of all three genes directly. Data are presented as dot plots with mean ± SEM; $n$ = 4–6 animals per group; statistical significance is assessed by 2-way ANOVA with Šídák's multiple comparisons test as a post-hoc test, **$p < 0.01$, *$p < 0.05$ vs an indicated group. Source data and the exact $p$-values are provided in a Source Data file. **c** The sequence of miR-483-5p from mouse, rat, man and guinea pig. **d** miR-483-5p target sequences located in 3'UTRs of the mouse, rat, human, chimp and rabbit *Pgap2* genes. Conserved nucleotides marked in red.

Mutations of the evolutionary conserved seed match sequences (Fig. 6a, d) in the 3'UTRs abolished miR-483-5p-mediated suppression, confirming that miR-483-5p affects all three target genes directly. This effect was further substantiated by a decrease in expression of *Pgap2*, *Gpx3* and *Macf1* genes in Neuro2a cells transfected with miR-483-5p (Fig. 4b).

Thus, we identified *Pgap2* as the most prominent candidate to mediate the effects of miR-483-5p in the amygdala. *Pgap2* was the only gene/protein out of twelve candidate molecules we examined that fulfilled all the essential criteria (stress-regulated expression, synaptosomal and membrane localisation, neuronal expression, and direct suppression by miR-483-5p) necessary to mediate the effects of miR-483-5p in the amygdala. Thus, *Pgap2* was selected as a primary miR-483-5p target for subsequent behavioural experiments.

**Anxiolytic and neuromorphological effects of miR-483-5p are mediated by suppression of *Pgap2***

Next, we established that stress-induced upregulation of miR-483-5p brings about a significant reduction of the *Pgap2* gene expression (Supplementary Fig. 5b), and that overexpression of miR-483-5p or suppression of the *Pgap2* gene resulted in the increase of the mature mushroom-shaped spines (Fig. 3b–h). Moreover, overexpression of both miR-483-5p and miR-483-5p-resistant *Pgap2* prevented the effects of miR-483-5p on spine morphology (Fig. 3g, h). These results strongly suggest that neuromorphological effects of miR-483-5p are caused by its interaction with *Pgap2*.

Stress-induced alterations in the amygdala gene expression pattern, dendritic arbour complexity and dendritic spine morphology are likely to coincide with the changes in the anxiety profile of

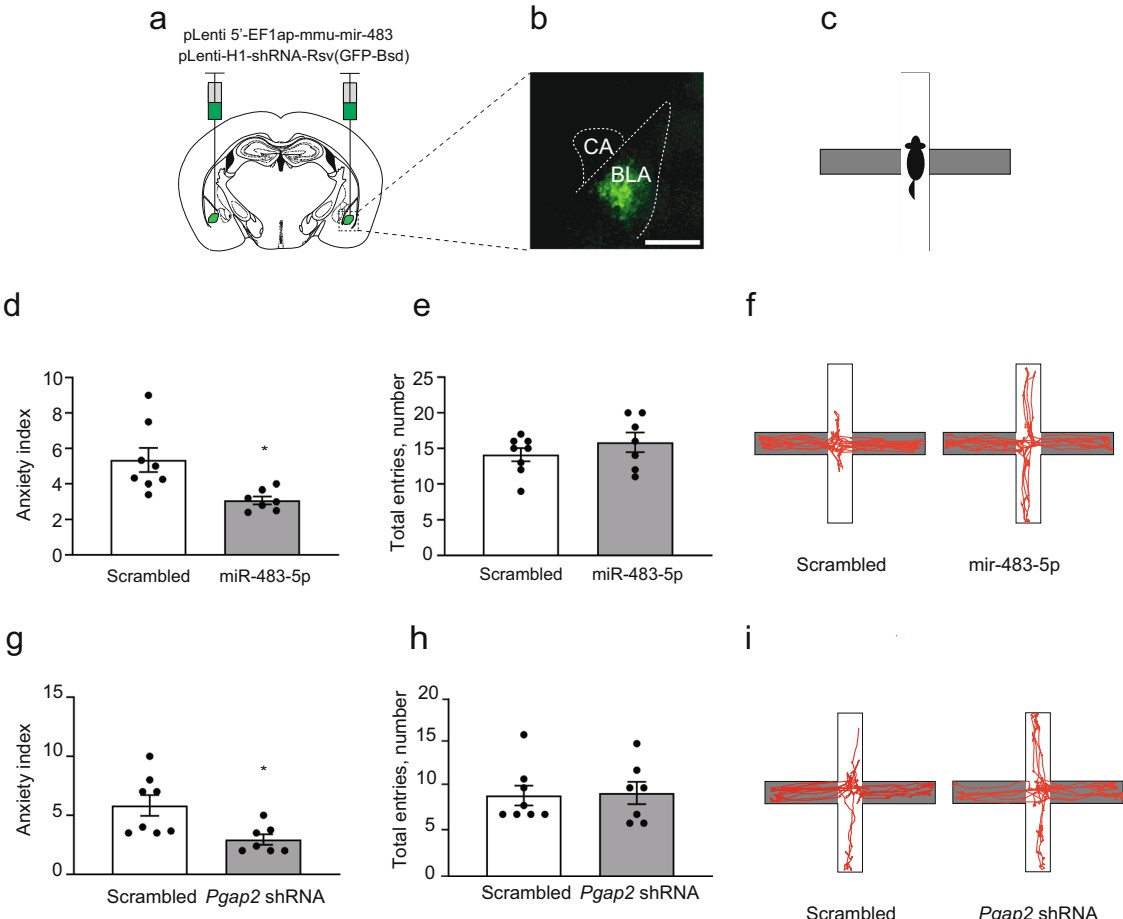

**Fig. 7 | Expression of miR-483-5p in the BLA in mice promotes anxiolysis through repression of *Pgap2* gene. a** Lentiviral particles containing the miR-483-5p sequence (pLenti-UbC-EGFP-mmu-miR-483-5p), or the control scrambled sequence, were injected bilaterally into the basolateral amygdalae (BLA) of mice. **b** Representative BLA injection site visualised with reporter protein GFP immunohistochemistry. Injection site was verified for all animals—see Supplementary Fig. 3 for the locations of injection sites. **c** After three weeks of recovery anxiety-like behaviour was tested in the elevated plus-maze. **d** Overexpression of miR-483-5p in the BLA led to decreased anxiety-like behaviour in mice compared to control animals as measured by the decrease in an anxiety index, a ratio between total arm entries to open arm entries. **e** The total number of arm entries (a measure of total locomotor activity) was not affected by the expression of miR-483-5p.

**f** Representative elevated-plus maze movements trajectories of mice injected with either miR-483-5p-containing or control lentiviral particles. **g** shRNA-mediated suppression of *Pgap2* (pLenti-H1-shRNA-Rsv(GFP-Bsd vector) in the BLA led to decreased levels of anxiety, thus recapitulating the behavioural effect of miR-483-5p. **h** Similarly to miR-483-5p, the total number of arm entries was not affected by the suppression of the *Pgap2* gene. **i** Representative elevated-plus maze movement trajectories of mice injected with lentiviruses containing either the *Pgap2*-supressing or control sequences. Data are presented as dot plots with mean ± SEM; *n* (scrambled) = 8, *n* (miR-483-5p & *Pgap2* shRNA) = 7 animals per group; statistical significance is assessed by unpaired two-sided *t* test, *$p < 0.05$ vs control (scrambled) group. Source data and the exact *p*-values are provided in a Source Data file.

animals[3,4,9,23,25]. Behavioural changes reflect a fine balance between stress-induced anxiogenic factors and the natural capacity of the brain to adapt to aversive stimuli. Thus, we next investigated how the overexpression of miR-483-5p in the mouse amygdala affects anxiogenesis and anxiety-like behaviour. Lentiviral particles expressing miR-483-5p were injected into the basolateral amygdalae of mice and then the animals were subsequently tested in the elevated-plus maze (Fig. 7a–c). Overexpression of miR-483-5p in the BLA resulted in an increase in the number of entries into the open arms of the maze by ~40%, the behaviour typical of low anxiety levels (Fig. 7d–f)[3,4,25]. Similarly, upon acute restraint stress, overexpression of the miR-483-5p prevented the development of the anxiety-like behaviour observed in the animals expressing the control virus (Fig. 8a–c). These results demonstrate that miR-483-5p is sufficient to both promote anxiolysis and prevent the development of anxiety-like behaviours.

miR-483-5p target gene *Pgap2* affects GPI-mediated anchoring of several enzymes, receptors and adhesion molecules in lipid rafts[26], but its role in regulating animal behaviour has never been examined. Thus,

we set out to determine whether the suppression of *Pgap2* recapitulates the behavioural consequences of overexpression of miR-483-5p. Basolateral amygdala neurons were transduced with lentiviral particles expressing *Pgap2*-targeting shRNA leading to a ~70% reduction of target gene expression (Supplementary Fig. 5c). Subsequently, elevated-plus maze testing revealed decreased anxiety-like behaviour in animals with reduced *Pgap2* expression as evidenced by a ~40% increase in the number of entries into the open arms of the maze (Fig. 7g–i), likewise observed in animals overexpressing the miR-483-5p (Fig. 7d–f). Similarly to miR-483-5p overexpression, *Pgap2*-targeted shRNA prevented the development of anxiety-like behaviour following stress (Fig. 8d–f). In keeping with the critical role of *Pgap2* in these phenomena, overexpression of miR-483-5p alongside miR-483-5p-resistant *Pgap2* prevented the effects of miR-483-5p on stress-induced behaviour (*Pgap2*[miR-483-5p] group, Fig. 8a).

The above results demonstrate that the neuromorphological and anxiolytic effects of miR-483-5p are caused by the suppression of its target gene, *Pgap2*.

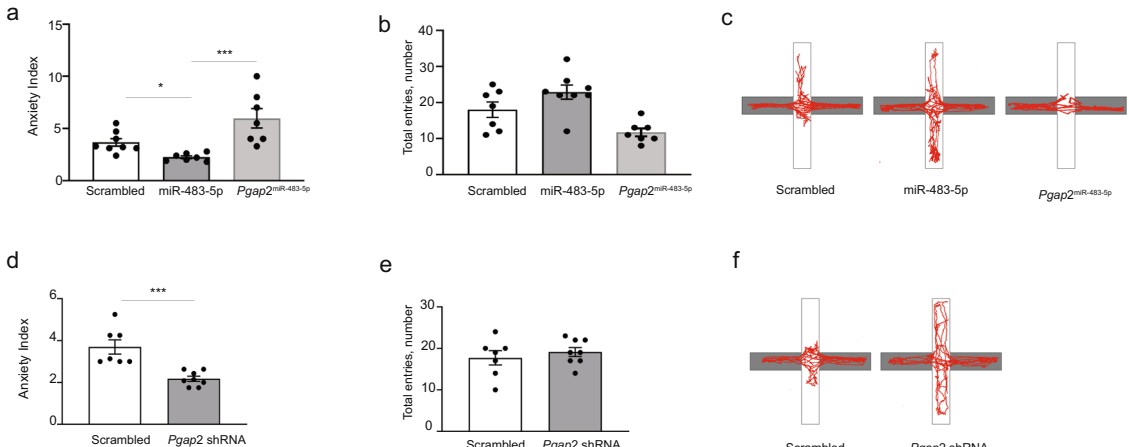

**Fig. 8 | Expression of miR-483-5p in the BLA prevents anxiogenesis through downregulation of *Pgap2* gene.** Lentiviral particles containing the miR-483-5p, *Pgap2*^miR-483-5p or the control scrambled sequence, were injected bilaterally into the basolateral amygdalae of mice. Following three weeks animals were subjected to 6-h immobilisation and next day tested on the elevated plus-maze.
**a** Overexpression of miR-483-5p in the BLA prevented development of the anxiety-like behaviour when compared with animals expressing scrambled, control sequence while co-expression of miR-resistant form of *Pgap2* with miR-483-5p reinstated anxiety-like behaviour to the level of control group. **b** The total number of arm entries (a measure of total locomotor activity) was not affected by the expression of miR-483-5p or *Pgap2*^miR-483-5p. **c** Representative elevated-plus maze movement trajectories of mice injected with lentiviral particles. **d** shRNA-mediated

suppression of *Pgap2* in the BLA led to decreased levels of anxiety-like behaviour in mice, thus recapitulating the behavioural effect of miR-483-5p. **e** Similarly to miR-483-5p, the total number of arm entries was not affected by the suppression of the *Pgap2* gene. **f** Representative movement trajectories of mice injected with lenti-viruses containing either the *Pgap2*-supressing or control sequences. Data are presented as dot plots with mean ± SEM; *n* (scrambled & *Pgap2*^miR-483-5p) = 7, *n* (miR-483-5p & *Pgap2* shRNA) = 8 animals per group; statistical significance is assessed by Kruskal–Wallis test with Dunn's multiple comparison test (**a**) or by a 1-way ANOVA with Bonferroni's multiple comparisons test (**b**) or by an unpaired two-sided *t* test (**d**, **e**), ***$p < 0.001$, *$p < 0.05$ vs control group. Source data and the exact *p*-values are provided in a Source Data file.

## Discussion

Stress can trigger the onset of a number of neuropsychiatric conditions that have their roots in an adverse combination of genetic and environmental factors[1,9,27]. While low levels of stress are counter-balanced by the natural capacity of the brain to adjust, severe or prolonged traumatic experience can overcome the protective mechanisms of stress resilience, leading to the development of pathological conditions such as psychotic states, depression and/or anxiety[1,9,27]. miRNAs have been identified as hub regulators of gene expression and have been implicated in a number of stress-related neuropsychiatric states[12,13]. Our understanding of the role of various types of miRNAs in shaping the transcriptome, and in the regulation of brain stress circuits leading to anxiety, is still in its infancy.

Here we found that miR-483-5p is enriched in the synaptic compartment of amygdala neurons upon psychological stress (Figs. 2, 3a). Long-term expression of miR-483-5p causes shrinkage of the distal segments of the dendritic arbour (Fig. 3b, c) and promotes the formation of mature, mushroom-like dendritic spines (Fig. 3d, e, g). Overexpression of miR-483-5p in the basolateral amygdala of mice is sufficient to exert an anxiolytic effect (Fig. 7d–f) and has a protective effect following acute psychological stress (Fig. 8a–c). miR-483-5p activity preventing expression of anxiety-like behaviour can be reca-pitulated by the knockdown of one of its target genes, *Pgap2* (Sup-plementary Figs. 5c, 7g–i, 8d–f). Both the miR-483-5p and its target sequence located in the 3'UTR of *Pgap2* coding mRNAs are conserved across various species (Fig. 6c, d). Thus, our studies demonstrate previously unknown miRNA-associated molecular events that control neuronal plasticity, offset stress-induced changes in dendritic mor-phology and promote anxiolysis.

The available literature on the role of miR-483-5p in the central nervous system is scarce. miR-483-5p was found to act as a tumour suppressor in gliomas[28], to regulate MeCP2 levels in foetal brains[29] and to regulate melatonin synthesis in the pineal gland[30], but there is no available literature on its role in the amygdala, hippocampus, cortex or other brain regions that collectively constitute the stress circuit.

Moreover, the role of miR-483-5p in any form of neuronal plasticity has never been shown before.

Changes occurring at synapses in response to experience are input-specific and often unique, thus cannot be explained by global regulation of translation, but rather require local mechanisms to reg-ulate protein synthesis[31]. These mechanisms are essential for translat-ing external cues into neuronal plasticity-related changes at the subcellular level, such as structural remodelling of dendritic arbour or dendritic spines[32]. miRNAs provide attractive instruments through which spatially restricted regulation of protein levels, leading to morphological remodelling, could be achieved. miRNAs have diverse expression patterns and can be restricted to or depleted from discrete cellular loci, such as dendritic spines[21,22]. We found that miR-483-5p is constitutively present and upregulated by stress preferentially in the synaptic compartment (Fig. 3a), thereby providing a mechanism for regulating protein synthesis locally. Such a precise spatiotemporal expression pattern is prerequisite to input-specific changes in den-dritic arbour and dendritic spine geometry[32].

The dendritic arbour architecture within the stress circuit, including the amygdala, determines how experience-related neuronal computations are conveyed and integrated[33]. Therefore, the molecular mechanisms of dendritic remodelling are key to our understanding of the development of stress-related neuropsychiatric disorders. miRNAs have recently been recognised as important players in the process of neuronal circuit adaptation to environmental stimuli. Specifically, miR-132, miR-124 and miR-134 were found to promote neurite outgrowth, dendritic maturation and activity-dependent dendritic remodelling by regulating small Rho GTPases[34–38]. Our study demonstrates that miR-483-5p causes selective shrinkage of the distal parts of the dendritic arbour and causes a small increase in dendritic spine density in vivo (Fig. 3g, h). Consequently, a less elaborate dendritic tree, which overall bears less synapses, has potentially major implications for the network properties of the anxiety circuit. This phenomenon has been demon-strated in the CA3 and CA1 regions of the hippocampus where stress causes shrinkage of basal dendrites leading to reversible cognitive

deficits[24,39]. On the other hand, it has been shown that stress promotes dendritic outgrowth in pyramidal neurons within the basolateral amygdala, which correlates with high anxiety[4]. Thus, stress-induced upregulation of miR-483-5p, with subsequent shrinkage of the amygdala dendritic arbour, positions this molecule in the pool of factors that counterbalance the adverse effects of stress.

Dendritic spines are microscopic protrusions from dendritic shafts that constitute the post-synaptic element of most excitatory synapses[40]. Consequently, the morphology of dendritic spines is a critical determinant of the efficacy of synaptic transmission. In addition to regulating synaptic plasticity, the geometry of spines can itself be bi-directionally modulated by synaptic activity, which can transform mushroom-like spines into thin spines and vice-versa[41]. Stressful experience often leads to changes in synaptic physiology resulting from changes in spine morphology[20,25]. Generally, spine shrinkage is associated with long-term depression, while spine enlargement correlates with long-term potentiation of synaptic transmission, the processes fundamental for learning and emotion[42,43]. miR-132[34,36,44], 134[45], 138[46] and 34[47] have been shown to regulate dendritic spine physiology and geometry. We found that overexpression of miR-483-5p or *Pgap2*-shRNA (mimicking its stress-induced upregulation and *Pgap2* mRNA downregulation, Supplementary Fig. 5) increases the percentage of mushroom-like spines with a simultaneous decrease in the number of filopodia and stubby forms of spines (Fig. 3). Filopodia and stubby spines are highly immature and rarely form synapses, whereas mushroom-like spines form strong, stable synapses with large pre- and post-synaptic specialisations intensely decorated with AMPA receptors[40,41]. Conversion of juvenile spines into established mushroom-like "memory" spines by miR-483-5p may provide a cellular substrate for stabilising the emotional status quo of the animal, making it more resilient to the detrimental effect of anxiogenic stimuli. Thus, miR-483-5p may be considered a molecular brake imposed on stress-induced dendritic spine plasticity in order to support anxiolytic behavioural profiles.

In keeping with the effect on dendritic arbour and spines, overexpression of miR-483-5p in the basolateral amygdala is indeed sufficient to promote anxiolytic behavioural states. Although other miRNAs have previously been implicated in regulating anxiety-like behaviour[16,18,19], only sole overexpression of miR-483-5p is sufficient to trigger anxiolysis as shown in the current study. This effect of miR-483-5p can be recapitulated by downregulating one of its target genes, *Pgap2*. The enzyme encoded by the *Pgap2* gene is involved in GPI-anchor maturation required for the stable expression of proteins at lipid rafts[26]. The GPI-anchored proteins (potentially affected by *Pgap2*) include hydrolytic enzymes (proteases), adhesion molecules (e.g. NCAM), membrane receptors (e.g. GDNF), and various regulatory proteins. The precise mechanism by which downregulation of *Pgap2* affects GPI-protein anchoring at dendritic spines, ultimately leading to an anxiolytic effect, requires further studies, but likely involves complex crosstalk between *Pgap2* and a number of its target molecules. The critical role of *Pgap2* in the regulation of the central nervous system functioning and structural remodelling is corroborated by the fact that mutations in this gene produce an autosomal recessive syndrome characterised by intellectual disability and mental retardation[48–50].

In summary, we identified miR-483-5p as a molecular brake that offsets stress-induced changes in dendritic and dendritic spine remodelling in the amygdala in order to promote anxiolysis. The behavioural outcome of overexpressing miR-483-5p can be recapitulated by the suppression of its target gene *Pgap2* – while neuromorphological and behavioural effects of miR-483-5p are prevented by simultaneous overexpression of miR-483-5p-resistant *Pgap2*. Thus, we identified and characterised previously unknown molecular events in the basolateral amygdala of mice sufficient to exert an anxiolytic effect. This finding may provide yet unexplored avenues for the development of anxiolytic therapies in humans.

## Methods

### Mice
Male wild-type mice (C57BL/6J) of 6–12 weeks of age were used. Three to five mice were housed in each cage with *ad libitum* access to food pellets and water in temperature-controlled rooms with 12-h light/dark cycles. All of the experiments were performed during the light half of the cycle. All procedures involving animals adhered to the Animals (Scientific Procedures) Act 1986 and Amendment Regulations 2012 as outlined in the UK law and approved by the University of Exeter Animal Welfare and Ethics Review Board, and the Institute of Pharmacology (Krakow, Poland) Local Ethical Committee in accordance with national and EU regulations.

### Behavioural analysis
Animals were left undisturbed in their home cages for one week before the experiments. Restraint stress and behavioural tests were performed during the light period of the circadian cycle as described[25]. Control animals were left undisturbed, and stressed animals were subjected to 6 h restraint stress sessions in a separate room. The mice were placed in their home cages in plastic restrainers with ventilation holes. Animals were left undisturbed for 12 h after the restraint stress before behavioural tests were conducted. The elevated-plus maze apparatus was made of two enclosed arms (50 × 10 × 30 cm) that formed a cross shape with the two open arms (50 × 10 cm). The maze was elevated 55 cm above the floor with two closed arms being dimly illuminated. Mice were placed individually on the central platform, facing an open arm, and allowed to explore the apparatus for 5 min. An overhead camera was used to record the activity. The videos were analysed semi-automatically using AnyMaze (Stoelting Europe) or Viewer II (Biobserve GmbH, Germany). After each mouse the maze was cleaned with 70% ethanol and left to dry for at least 2 min before introducing the next animal to the maze. Behavioural analyses were performed blindly. An anxiety index was calculated by the number of total arm entries divided by open arm entries. Locomotor activity was measured by the total number of entries to all arms of the maze.

### Microarray study
Immediately after stress, wild-type control ($n = 12$) and restraint-stressed mice ($n = 12$) were anaesthetised, intraperitoneally using sodium pentobarbital, and transcardially perfused with ice-cold phosphate buffered saline (PBS). Amygdalae were isolated using a dissecting microscope in ice-cold ACSF (Glucose 25 mM, NaCl 115 mM, NaH2PO4·H2O 1.2 mM, KCl 3.3 mM, CaCl2 2 mM, MgSO4 1 mM, NaHCO3 25.5 mM, pH 7.4) and stored in RNAlater (Qiagen) at −20 °C until further processing. Total RNA enriched in small RNA was extracted using mirVANA kit (Applied Biosystems). RNA quantity and quality (A260/280 > 1.80) were measured using NanoDrop ND-1000 Spectrophotometer (Nanodrop Technologies). RNA samples were sent to Febit Biomed Gmbh (Germany) for miRNA expression profiling. RNA samples were analysed with the Geniom Realtime Analyzer (Febit Biomed Gmbh) using the Geniom Biochip MPEA mus musculus (Febit Biomed Gmbh) containing 7 replicates of reverse complement probes for all mature miRNAs and mature sequences as annotated in the Sanger miRBase version 14.0 September 2009. RNA pulled from three mice was hybridised to one replicate giving a total n number of 4 samples per group. Briefly, samples were labelled with biotin using microfluidic-based enzymatic on-chip labelling of miRNAs[51]. Hybridisation was performed automatically for 16 h at 42 °C followed by automatic biochip washing and signal measurement. The signal enhancement process was combined with streptavidin-phycoerythrin detection of biotin to maximise signal sensitivity. Array images were analysed using the Geniom Wizard software (Febit Biomed Gmbh). For each array, the median signal intensity was extracted from the raw data file such that for each miRNA seven intensity values have been calculated corresponding to each replicate copy of miRNA-Base on the

array. After background correction, median values were calculated from the seven replicate intensity values of each miRNA. Microarray data were analysed using R's LIMMA package.

## Cell culture experiments

Neuro-2A or HEK cells were incubated at 37 °C at 5% $CO_2$ in DMEM culture media with 1% penicillin-streptomycin, 1% non-essential amino acids and 5% foetal bovine serum. Experiments were performed when the cultures reached 70–80% confluence. The cells were transfected using either a control plasmid containing a scramble sequence or pEGP-miR483-5p-expressing plasmid (Cell Biolabs). pEGP-miR483-5p was modified by replacing the EGFP sequence with mCherry. 24 h after transfection cells were treated with 1 µM dexamethasone for 3 constitutive days, followed by RNA extraction.

## Primary mouse amygdala neuronal culture preparation

Primary amygdala neuronal cultures were prepared as described[20]. Amygdale from P1C57BL6J wild-type mice were dissected and placed in a dish containing 9.1 mM glucose, 25 mM Hepes, 5 mM KCl, and 120 mM NaCl. Tissue was digested (5 mg of pronase E and 5 mg of thermolysin; Sigma) in 10 mL of the buffer for 30 min at RT, triturated, and plated on poly–D-lysine (Sigma)-coated coverslips. After 24 h, 5 μM cytosine β–D-arabinofuranoside (Ara-C; Sigma) was added for 48 h. Neurons were maintained for 17–21 days in vitro (DIV) in Neurobasal medium with supplements at 37 °C in a humidified atmosphere of 5% $CO_2$/95% air. Lipofectamine was used to transfect the neurons at day 7–9 with plasmid overexpressing miR-483-5p under the CMV promoter.

## Dendritic spine and Sholl analyses

Spine morphology was assessed in primary mouse amygdala neurons according to previously published criteria[20,25]: mushroom spines– <2 µm in length, >0.5 µm in width, and connected to the dendritic shaft by a narrower portion (neck); stubby spines–<2 µm in length, >0.5 µm in width, and lack of a defined neck; thin spines–<2 µm in length, <0.5 µm in, and with a neck; filopodia–>2 µm in length, <0.5 µm in width, without a distinct spine head, and irregular spines with more than one neck and/or head. In order to perform Sholl analysis, the whole dendritic tree of the neurons was digitally traced using the Imaris software. Concentric circles were then drawn in 10 µm intervals, starting from the centre of the soma and the number of intersections of dendritic branches with each circle was measured. A total of 55 neurons were analysed (21 transfected with the scrambled sequence and 34 with the mir-483-5-expressing vector).

## Synaptosomes preparation

Synaptosomes were extracted using Syn-PER™ reagent (Thermo Scientific), following manufacturer protocol. Briefly, animals were anaesthetised with pentobarbital and then perfused with RNase free, ice-cold PBS. Amygdalae were dissected and weighed. Tissue samples were homogenised in 10 volumes of the Syn-PER™ reagent followed by centrifugation (1200 × $g$ for 10 min at 4 °C). The pellets containing cell debris were discarded and synaptosomal fraction was concentrated by subsequent centrifugation (15,000 × $g$ for 20 min at 4 °C). Resulting highly enriched synaptosomal fractions were used for RNA extractions.

## Quantitative RT-PCRs

Animals were anaesthetised with pentobarbital and then perfused with RNase free, ice-cold PBS. Amygdalae were extracted from a thick coronal slice (−0.58 to −2.3 mm relative to Bregma) and submerged in RNalater solution (Qiagen) to prevent RNA degradation. RNA was extracted using RNeasy Lipid tissue mini kit (Qiagen) accordingly to the manufacture protocol. 1–2 µg of RNA was converted to cDNA using Superscript III (Invitrogen). RNAs from cultured cell lines were

extracted using RNeasy Mini Kit (Qiagen) and converted to cDNA as described above. miR subfraction of the RNA from whole amygdalae tissue and/or synaptosomal fraction was purified using miRNeasy Mini Kit (Qiagen) and miScript II RT Kit (Quaigen) was used to convert mature miRNA into cDNA. miScript SYBR® Green PCR Kit (Qiagen) was next used to analyse the relative levels of miR-483-5p using the custom synthesised primers for mir-483-5p accordingly to the miScript Primer Assay system (Quiagen, Cat#: 218300). Levels of miR-483-5p were normalised to Rnu6 nuclear RNA.

The qPCR conditions for relative mRNA quantitation started with 15 min denaturation step at 95 °C followed by 40 cycles of 95 °C−15 s, 55 °C−30 s, 72 °C−30 s. Following additional 10 min of incubation at 72 °C. mRNA levels were quantified and normalised to the ß-actin coding mRNA (Forw:5'-TGCTCCTCCTGAGCGCAAGTACTC, Rev:5'-CGGACTCATCGTACTCCTGCTTGC).

## Immunohistochemistry

Immunohistochemistry was performed as described previously[2]. Briefly, animals were anaesthetised with pentobarbital and then perfused with with ice-cold PBS followed by 4% PFA in PBS. The brains were extracted and fixed in 4% PFA in PBS overnight at 4 °C. The following day, brains were washed with PBS and 70 µm thick coronal sections were cut using a vibratome (Campden Instruments, UK). Sections were blocked with 10% FBS in PBS-T (PBS + 0.1% Triton X-100) for 1 h at RT followed by incubation with primary antibodies overnight. Antibodies used were: PGAP2 (Abcam, 1:500), GPX3 (Abcam, 1:1000), MACF1 (Abcam, 1:1000), NeuN (Chemicon, 1:1000), GFAP (Abcam, 1:2000), HOMER (Synaptic Systems, 1:500). The secondary, Alexa-Fluor conjugated antibodies were chosen accordingly to wave length emission (Molecular Probes). Sections were mounted with Fluorsave medium (Calbiochem) and photographs were taken using a Zeiss LSM 5 Exciter confocal microscope.

## DiI Labelling

The brains were fixed (1.5% paraformaldehyde for 1 h at room temperature) and 170 µm coronal slices containing amygdalae were cut on a vibrating microtome. Small DiI crystals (Molecular Probes) were applied onto GFAP expressing neuronal cell bodies for 24 h. After further fixation, the sections were mounted and the dendritic spines were visualised using a Zeiss LSM5 Exciter or Leica sp5 and analysed.

## Fluorescent in situ hybridisation (FISH) followed by immunohistochemistry

Mouse brains extracted from the stress-subjected animals were fixed overnight at 4 °C in 4% PFA in PBS. The following day, brains were washed in DEPC-treated PBS at room temperature and cut as described above. The coronal sections were placed on polysine glass microscope slides (VWR) and left to dry. Dried sections were stored at -80 degrees until use. Directly before use, the slides were submerged in DEPC-PBS to remove the PBS precipitate. In the meantime digoxigenin (DIG)-labelled LNA probes targeting miR-483-5p as well as the control, scramble sequences were denatured in hybridisation buffer (50% v/v deionized formamide, 0.2 M NaCl, 50 mM EDTA, 10 mM Tris-HCl, pH 7.5, 5 mM $NaH_2PO_4 \cdot H_2O_2$, 5 mM $Na_2HPO_4$, 0.05 mg/ml tRNA from baker's yeast) for 5 min at 70 °C and placed onto slides. The dilution of the probe in the hybridisation buffer generating the best signal-to-noise ratio was established empirically as 1:1000. Sections were covered with cover glasses and the hybridisation was performed at 65 °C overnight in a chamber humidified with 50% v/v formamide containing 1× SSC buffer. The next day sections were washed three times at 65 °C for 30 min each in wash solution (50% v/v formamide, 1× SSC, 0.1% Tween 20), followed by two washes for 30 min each in 1× MABT (100 mM maleic acid, 150 mM NaCl, pH 7.5, 0.1% Tween 20) at RT. The sections were blocked with blocking buffer (0.5% blocking reagent [Roche] dissolved in TNT buffer [0.1 M Tris-HCl pH7.5, 0.15 M NaCl,

0.05% Tween-20]) for 1 h at RT. Anti-Digoxigenin-POD (Roche) antibody at 1:200 dilution in blocking buffer was applied onto the sections and incubated for 1 h at RT, followed by three washes with PBST (PBS containing 0.1% Triton X-100). The biotin deposition was performed using TSA Plus Biotin Kit (Perkin Elmer), followed by three subsequent washes with PBST and 45 min incubation with ABC reagent (Vector Laboratories). Then, fluorescein was deposited using TSA Plus Fluorescein Kit (Perkin Elmer). For immunodetection of NeuN, PGAP2 or Homer after in situ hybridisation, the sections were treated as described above, following blocking with 10% FBS in PBS-T (PBS + 0.1% Triton X-100) for 1 h at RT and incubation with primary antibodies overnight. The next day the bound primary antibodies were detected using fluorescently labelled secondary antibodies.

## Dual-Glo luciferase activity

To examine the direct interaction of miR-483-5p and *Pgap2*, *Gpx3* and *Macf1* 3′UTR regions we used the Dual-Glo® Luciferase assay (Promega) in conjunction with pmiRGLO Dual-Luciferase vector. Briefly, the 3′ UTR regions containing the miRNA-target sequences for *Pgap2*, *Gpx3* or *Macf1* genes were cloned into the MCS of pmiRGLO Dual-Luciferase vector at 3′ end of the Firefly luciferase gene (*luc2*) using SacI and XbaI restriction sites. The decrease of luciferase mRNA/protein activity upon expression of miR-483-5p (or control, scrambled sequence) cloned in vector and its binding to target sequences was performed in Neuro-2a cells and measured with PHERAStar® Plus (BMG LABTECH). The specificity of miR-483-5p target sequence interaction was verified by site-directed mutagenesis introducing the point mutations abolishing/reducing miRNA-target sequence interaction. The activity of miR-483-5p was calculated as a ratio of Firefly luminescence to *Renilla* luminescence for the scramble and miR-483-5p expression. Data is represented in the graphs as a percentage of decrease relative to control, scramble sequence.

## Stereotaxic surgery

Stereotaxic surgery was performed under stereotaxic guidance using a stereotaxic frame (Stoelting, USA) and aseptic techniques as described in ref. 2. Mice aged 6-8 weeks old were used for the surgery. The animals were treated with 5% isoflurane with 4 l/min of oxygen to induce anaesthesia. The mouse heads were immobilised and placed in the stereotaxic frame (Stoelting Europe) while supplied with a constant rate of 2.5% isoflurane and 1 l/min oxygen through a mouse face mask (Stoelting Europe). The stereotaxic coordinates used to target the basolateral amygdala (BLA) were corrected for each experimental group, and were; -1.5 mm antero-posterior (AP), ±3.0 mm to ±3.4 mm medio-lateral (ML) and −3.5 mm to −4.2 mm and −4 mm to −4.7 mm dorsoventral (DV). First, the required amount of lentiviral particles was aspirated into a 10 μl NanoFil© (World Precision Instruments) syringe. The needle (33-gauge) was inserted to −4 mm to −4.7 mm (DV) and the lentiviral particles (0.3 μl) were injected at a rate of 100 nl per minute using an UMP-3.1 micropump (World Precision Instruments). After the injection, the needle was left undisturbed at the location for another 5 min after which it was retracted back to −3.5 mm – −4.2 mm and another 0.3 μl injected. After a 5 min resting period, the needle was completely withdrawn from the brain. Lentiviral particles were injected bilatrally. Following the injection, the incision wound was sutured or closed using Vetbond, and animals were given appropriate analgetics for at least 3 days. Animals were left to recover altogether for 2–4 weeks before behavioural experiemnts.

## Lentiviral particles

miR-483-5p sequences from pEGP-mir483-5p, cDNA of miR-483-5p-resistant *Pgap*2 co-expressing miR-483-5p, or a control scramble sequences were cloned into LV-pUltra plasmid (Addgene #24129) and, co-transfected with pCMV-delta R8.2 (Addgene #12263) and pCMV-VSV-G (Addgene #8454) in HEK293T cells in order to generate lentiviral particles. The viral particles were purified and concentrated by ultracentrifugation. The estimated lentiviral titre was $9.9 \times 10^7$ particles/ml. Lentiviral particles to knockdown transcripts of *Pgap2* (NM_001291358.1 & NM_145583.3) and control viruses expressing the scrambled sequence (GTCTCCACGCGCAGTACATTT) were purchased from AMSBIO. The titre was estimated to be $1.07 \times 10^7$ particles/ml.

## Statistics

Data are presented as dot plots with mean ± SEM unless stated otherwise in figure legend. Statistical Student's *t* test (when two groups were compared) or analysis of variance (ANOVA) followed by Tukeys HSD Multiple Comparison post-test were used as appropriate. *P* values of less than 0.05 were considered significant (reported as *P* for ANOVA and *p* for the post-test).

## Reporting summary

Further information on research design is available in the Nature Portfolio Reporting Summary linked to this article.

## Data availability

All data generated or analysed during this work are included in this article and its Supplementary Information files. Source data are provided in this paper. The microarray datasets are available in the Gene Expression Omnibus (GEO) under the accession code GSE227028. Source data are provided in this paper.

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

## Acknowledgements

The study was supported by the Marie Sklodowska Curie ITN "Extra-brain" (ID 606950), The Leverhulme Trust and the Cleopatra Trust grants to Robert P, Polish National Science Centre Grant number 2013/08/A/NZ3/00848 to Ryszard P, the AMS Springboard award SBF005\1102 and the MRC Career Development Award MR/T031115/1 to VM.

## Author contributions

M.M.—Conceptualization, Methodology, Formal analysis, Investigation, Data Curation, Writing—Review & Editing, Visualisation, Supervision, Project administration; A.E.S.—Conceptualization, Methodology, Formal analysis, Investigation, Data Curation, Visualisation, Project administration; J.B.K.—Conceptualization, Methodology, Formal analysis, Investigation, Data Curation, Visualisation, Project administration; V.B.—Investigation, Formal analysis; S.P.—Software, Formal analysis, Investigation, Data Curation; A.L.R.—Investigation, Formal analysis; L.K.—Investigation, Formal analysis; K.M.—Investigation, Formal analysis, Writing—Review & Editing; N.S.—Software, Formal analysis, Investigation, Data Curation; V.D.P.—Investigation, Formal analysis; A.K.—Investigation, Formal analysis; Ryszard P.—Conceptualization, Resources, Data Curation, Supervision, Project administration, Funding acquisition; V.M.—Conceptualization, Methodology, Formal analysis, Investigation, Resources, Data Curation, Writing—Review & Editing, Visualisation, Supervision, Project administration, Funding acquisition; Robert P.—Conceptualization, Formal analysis, Resources, Data Curation, Writing—Original Draft, Writing—Review & Editing, Visualisation, Supervision, Project administration, Funding acquisition.

## Competing interests

The authors declare no competing interests.

## Additional information

[1]University of Exeter Medical School, Department of Clinical and Biomedical Sciences, University of Exeter, Exeter, UK. [2]Pharmacy Department, Alberta Health Services, Calgary, AB, Canada. [3]Department of Molecular Neuropharmacology, Institute of Pharmacology, Polish Academy of Sciences, Krakow, Poland. [4]UK Dementia Research Institute at Imperial College London, London, UK. [5]Department of Brain Sciences, Imperial College London, London, UK. [6]Department of Clinical Laboratory Diagnostics, Medical University of Bialystok, Bialystok, Poland. [7]Nencki Institute of Experimental Biology, Polish Academy of Sciences, Warszawa, Poland. [8]School of Physiology, Pharmacology and Neuroscience, Faculty of Life Sciences, University of Bristol, Bristol, UK. [9]These authors contributed equally: Mariusz Mucha, Anna E. Skrzypiec, Jaison B. Kolenchery. ✉e-mail: mwmucha@gmail.com; valentina.mosienko@bristol.ac.uk

