## [Peer Review File · Nature Communications]

miR-483-5p offsets functional and behavioural effects of stress through synapse-targeted repression of Pgap2 in the basolateral amygdalaReviewers' comments:

Reviewer #1 (Remarks to the Author):

In this beautifully and clearly written research paper Skrzypiec et al. identify a novel subset of microRNAs (miRNAs) that are upregulated in the basolateral amygdala (BLA) by exposure to acute stress. The authors focus their study on miR-483-5P and show it is enriched in synaptosomes isolated from the BLA. Next, the authors show that up regulating of miR-483-5P in amygdala culture decreases the dendritic tree and changes in proportion of mature vs immature dendritic spines. Using a series of bioinformatic, in vitro and in vivo experiments, the authors narrow down a list of potential direct targets that miR-483-5P may act on to counter the stress response, and identify Pgap2. Taken together, this study is well executed, and the results are presented clearly. However, the conceptual novelty of the findings is not very high, and there is insufficient data presented to support some of the main conclusion and statements of the study. This makes this manuscript insufficient for publication in a journal like Nature Communications.

Major comments

1. As the authors acknowledge in the text, there are already publications showing a role for amygdala microRNAs in the response both to acute and chronic stress using bulk tissue profiling. To my knowledge, there are no studies published focusing on the role of synaptic miRNAs in stress, which is the focus of this paper. However, in order to best study this question the authors should have run the screen described in figure 1 on isolated synaptosomes rather than a crude microdissection of the amygdala. Such an analysis would have likely give very different result.
2. For the paper to be relevant for human anxiety, the authors should present conservation data from mouse to human for both miR-483-5P and its bioinformatically predicted targets. For example, I have run a quick analysis in TargetScanMouse 7.1, and found that miR-483-5P is classified as, "Poorly conserved but confidently annotated miRNA Families". Furthermore, the seed match sequence for GPX3 is conserved only in mouse and rat, but not human. Additionally, I could not find the predicted miR-483-5P binding in mouse 3'UTR of the other top candidates.
3. To conclude that that miR-483-5P offset the effects of stress as the title of this paper claims, a crucial experiment is missing. MiR-483-5P should be overexpressed in stressed animal and this should have a stress protective effect. Alternatively, knocking down miR-483-5P in vivo, should have a anxiogenic effect.
4. To conclude that miR-483-5P acts in neurons to regulate dendritic branching and spines, the authors should give direct evidence that miR-483-5P is endogenously expressed in amygdala neurons. This can be done by ISH for the miR-483-5P along with ISH or IHC for a neuronal marker.

Minor comments

1. Some of the experiments performed in cell lines or primary culture could have been easily done in the whole animal, which would give much more physiological relevance to the authors' conclusions. For example, testing the effects of stress and miR-483-5P on dendritic branches and spines in slices from mice amygdala instead of in primary culture would teach us more about the role of this miR in stress-induced anxiety. Similarly, the regulation of the putative target genes by miR-483-5P and stress could have been tested in mouse amygdala tissue.
2. Need to add an in vivo validation of the overexpression of miR-483-5P that is described in figure 6. It should also be shown in vivo that this leads to down regulation of Pgap2. Similarly, the shRNA to Pgap2 experiments in missing in vivo validation.
3. The authors should include the full list of stress-regulated miRNAs in the supplemental material.
4. Figure 5 a, looks like a typo: is Macf1 3'UTR is 22,074 pb?
5. Why was a different amount of nucleotides (1-3) mutated in the different 3' UTR tested in figure 5?
6. The authors should have used more than one behavioral test as a phenotypic readout of anxiety and stress.

Reviewer #2 (Remarks to the Author):

This study is aimed at assessing whether the effects of restraint stress on neuronal structure in the basolateral amygdala and on anxiety-like behavior are mediated by selective microRNA regulation of synaptic genes. Microarray analysis and qPCR experiments were combined to identify candidate microRNAs in the BLA amygdala that are regulated by 6 h restraint stress in mice. Microarray analysis revealed stress-induced upregulation of 5 microRNAs, but the study focused on miR-483-5p because it exhibited the highest increase. Interestingly, stress-induced upregulation of miR-483-5p was localized to the synaptosomal, but not cytosol, fraction. Furthermore, miR-483-5p overexpression in primary mouse amygdala neurons induced shrinkage of their distal dendritic arbor, but an increase in the proportion of mature versus immature spines. To identify potential target genes of miR-483-5p, bioinformatics analysis was conducted. Out of twelve potential targets genes identified by cross-referencing 3 databases, Pgap2, Gpx3, and Macf1 were found to be upregulated by dexamethasone in Neuro2a cells. This effect was reduced by miR-483-5p overexpression. Finally, overexpressing miR-483-5p or silencing Pgap2 directly into the BLA amygdala of mice increased the number of open arm entries in the elevated-plus maze test. Based on these results, the authors concluded that miR-483-5p offsets the functional and behavioral effects of stress via repression of Pgap2 in the synaptic compartment of BLA neurons.

The idea of searching for genes and their microRNA regulators that are potentially involved in stress-induced functional changes in the BLA amygdala at the level of the synapse is novel and promising. The multidisciplinary approach employed in the study is a strength. However, I have the following concerns:

Overall:

1. There is a lack of information regarding the results of the statistical analyses for all the data presented in the manuscript. Only p values are mentioned. The only exception is for the data shown in Figure 2a, where an F value with its corresponding degrees of freedom is provided. Without this information, it is not possible to determine the validity of the findings presented.
2. An experiment demonstrating coexpression of miR-483-5p and Pgap2 in BLA neurons of adult mice is missing.
3. The main conclusion of the study is that “miR-483-5p offsets functional and behavioural effects of stress through synapse-targeted repression of Pgap2 in the basolateral amygdala”. However, this conclusion is not accurate because:
 - first, the study does not provide (in vivo) evidence of (a) alterations in dendritic and/or spine structure in BLA neurons of mice subjected to restraint stress and (b) of attenuation of these structural changes by miR-483-5p over-expression in BLA neurons
 - second, the study does not provide (in vivo) evidence that miR-483-5p overexpression in BLA neurons of mice downregulates Pgap2 expression
4. In vivo confirmation of overexpression of miR-483-5p and of downregulation of Pgap2 in BLA amygdala neurons in the lentiviral experiments is missing. These changes were only tested in vitro in a separate assay
5. The lentiviral manipulations were not designed to target amygdala neurons selectively

Specific points:

Statistical analysis:

- The sample size for the qPCR experiments (n=3) is low
- data should be plotted as individual values

Abstract:

- It is not clear if the authors are referring to humans and/or rodents
- the following statements are not supported by the results (as explained in the overall comments):
“Here we show that miR-483-5p in the amygdala counterbalances the structural, functional and behavioural consequences of psychological stress to promote anxiolysis” and

“Our results demonstrate that miR-483-5p is sufficient to confer an anxiolytic effect and point to miR-483-5p-mediated repression of Pgap2 as a critical cellular event offsetting the functional and behavioural consequences of psychological stress”.

Introduction:

- The references cited do not always correspond to the text preceding them (for instance see the first sentence)
- The authors state that “The molecular functioning of central neurons is orchestrated by small, noncoding RNA sequences called microRNAs”. However, not only microRNAs orchestrate this function.
- It is not clear what “pathway fine-tuning” refers to and how exactly miRNAs contribute to this.
- The authors need to specify the species they are referring to (rodents and/or humans)
- It is mentioned that the molecular and cellular mechanisms by which microRNAs “regulate stress resilience are largely unknown”. It seems to me that the same is the case for stress susceptibility.
- The introduction does not provide a clear rationale for the study and does not mention the study’s goal

Results:

- the sex, age, and strain of mice needs to be mention right from the start. This is only mentioned in the method sections, at the end of the manuscript
- Regarding the bioinformatics approach, more detail is needed about the criteria that was used to identify potential gene targets. For instance, it is not clear how exactly “stress-related” targets were identified.
- Are there genes that are not altered after dexamethasone exposure in the in vitro studies?
- Regarding the immunofluorescence image, the quality is low and critical information is missing, including antibody specificity and phenotype of labelled cells.
- Fig 4: without showing the precise statistical analysis and the individual data, it is difficult to determine whether stress alters the expression of Gpx3 and Macf1 in the synaptosomal fraction

Reviewer #3 (Remarks to the Author):

Review of: “miR-483-5p offsets functional and behavioural effects of stress through synapse targeted repression of Pgap2 in the basolateral amygdala”

This is a mostly well-written, interesting, and potentially important paper, providing novel evidence for a miRNA, miR-483-5p, in amygdala-dependent regulation of the stress response. The authors argue that ‘miR-483-5p is a molecular brake that offsets stress-induced changes in dendritic and dendritic spine remodelling in the amygdala in order to promote anxiolysis, and that the behavioural outcome of overexpressing miR-483-5p can be recapitulated by the suppression of its target gene Pgap2.’ While the experiments are elegant and the data are interesting, and mostly compelling, I have a number of remaining concerns about the experiments outlined and the logic of the interpretation of the experimental results, as outlined below.

Major Concerns:

- Did the 5 mRNAs that are mentioned from the discovery microarray have a level of significance that survived correction for multiple testing? Is the $P < 0.001$ level the nominal significance level or the p-value after correction?
- After validation, the authors state, “Therefore, our subsequent studies focused on the role of amygdalar miR-483-5p in the stress response and on its mechanism of action.” But it is not clear why they focused on miR-483-5p over the 4 other candidates that were most differentially expressed, and

appeared to all similarly survive validation.

- Figure 2e – example figures of the dendritic change: the level of contrast / brightness / sharpness of the two figures looks very different. Specifically the miRNA transfected example looks blurry and a simple interpretation of the quantitative result is that if they were not equivalent images, the miRNA may appear to have more mushroom-like and fewer filopodia, but that may be because one can't visualize the filopodia with blurry pictures.

- Does the shRNA suppression of Pgap2 lead to the same cellular / dendritic / spine changes that overexpression of miR-483-5p does?

- Authors show that miR-483-5p is significantly upregulated following restraint stress but then go on to argue that it mediates a compensatory / resilient process via Pgap2, and that suppressing Pgap2, or overexpressing miR-483-5p, is anxiolytic on a plus maze. However, the logic is not complete, in that they identified the miR following restraint stress, and generally stress is going to increase a number of anxiety / fear / stress behaviors, but they show that the miR is anxiolytic. To argue that it is a compensatory process, it would seem important to show that either inhibiting miR-483-5p (e.g. with a miR sponge) or overexpressing miR-483-5p would increase or decrease, respectively, restraint stress-dependent behaviors, based on the initial observation.

Minor Concerns:

- It is not clear how long after the restraint stress that animals were sacrificed for miRNA analysis

- In the first section of results, the authors state, "We then observed that stress caused a further 4-fold increase of miR-483-5p in the synaptosomal fraction, but not in the cytosol (Figure 2a, $F(3, 11) = 23.11$, $P < 0.00005$, $p < 0.001$, $n = 3-4$ per group)." – it is not clear why two different p-values are presented here.

- There were several typographical errors throughout; I encourage a careful rereading / editing for typographical errors and grammar.

Dear Sir/Madam

Please find below our comments addressing the Reviewers' concerns regarding our manuscript "**miR-483-5p offsets functional and behavioural effects of stress through synapse-targeted repression of *Pgap2* in the basolateral amygdala**"

Reviewers' comments:

Reviewer #1:

Major comments

Question. As the authors acknowledge in the text, there are already publications showing a role for amygdala microRNAs in the response both to acute and chronic stress using bulk tissue profiling. To my knowledge, there are no studies published focusing on the role of synaptic miRNAs in stress, which is the focus of this paper. However, in order to best study this question the authors should have run the screen described in figure 1 on isolated synaptosomes rather than a crude microdissection of the amygdala. Such an analysis would have likely give very different result.

Answer: The microRNA profiling study (microarray analysis of the microRNAs affected by stress) allowed us to identify and functionally characterise the microRNA molecules previously unknown to be involved in anxiodepressive disorders. We did not assume nor predict a priori the need for synaptic microRNA profiling as its subcompartmental localisation and function we discovered, stemmed from the results of our experiments we performed post-microarray analysis. We have subsequently used a broad array of methods, including synaptosomal preparations to confirm the role of the newly discovered role of miR-483-5p in the synaptic compartment. Indeed, we agree that focusing on the synaptosomal population of microRNAs in the initial microarray screening could potentially be an interesting avenue to pursue in the future.

Question. For the paper to be relevant for human anxiety, the authors should present conservation data from mouse to human for both miR-483-5P and its bioinformatically predicted targets. For example, I have run a quick analysis in TargetScanMouse 7.1, and found that miR-483-5P is classified as, "Poorly conserved but confidently annotated miRNA Families". Furthermore, the seed match sequence for GPX3 is conserved only in mouse and rat, but not human. Additionally, I could not find the predicted miR-483-5P binding in mouse 3'UTR of the other top candidates.

Answer: We have now performed detailed analyses suggested by the Reviewer and found that miR-483-5p shares homology with both human and mouse mir-483-5p

sequences. Only 2 out of 21bp are different and those differences are not within the seed region of our primary target, Pgap2 (as demonstrated in Fig 6C). Also, the miR-483-5p target sequence in the mRNA coding for the Pgap2 gene in various species is highly conserved (new Fig 6D). The algorithms that predict the target genes tend to change over time based on new bioinformatics, algorithm modifications and new experimental data. Our study, apart from the *in-silico* studies, provided several independent experimental lines of evidence for the miR-483-5P directly interacting at the 3' UTR of the genes we have studied.

Question. To conclude that that miR-483-5P offset the effects of stress as the title of this paper claims, a crucial experiment is missing. MiR-483-5P should be overexpressed in stressed animal and this should have a stress protective effect. Alternatively, knocking down miR-483-5P in vivo, should have a anxiogenic effect.

Answer: The requested experiments have now been included in Figure 8 and discussed on page 7. We found that indeed, overexpression of miR-483-5P in mouse amygdala prevented stress-induced anxiety. Our observation was additionally complemented with the expression of shRNA targeting miR-483-5p target gene Pgap2 in mouse amygdala, demonstrating that knocking down this target gene (either with or without stress; new Fig 7g, 8d) conferred the same behavioural phenotype.

Question. To conclude that miR-483-5P acts in neurons to regulate dendritic branching and spines, the authors should give direct evidence that miR-483-5P is endogenously expressed in amygdala neurons. This can be done by ISH for the miR-483-5P along with ISH or IHC for a neuronal marker.

Answer. The requested experiments have now been included in Figure 2a (and included in text page 3) where we demonstrate the expression of the miR-483-5P by amygdala neurons and its co-localisation with the synaptic marker HOMER. We found colocalization of the miR-483-5P with HOMER in a subset of amygdala synapses. In addition, we have performed another set of new experiments demonstrating that, similarly to miR-483-5p, its target Pgap2 also co-localizes with neurons and is present in amygdala synapses (new Fig 2b)

Minor comments

Question. Some of the experiments performed in cell lines or primary culture could

have been easily done in the whole animal, which would give much more physiological relevance to the authors' conclusions. For example, testing the effects of stress and miR-483-5P on dendritic branches and spines in slices from mice amygdala instead of in primary culture would teach us more about the role of this miR in stress-induced anxiety. Similarly, the regulation of the putative target genes by miR-483-5P and stress could have been tested in mouse amygdala tissue.

Answer. We have now performed the requested dendritic spine density and morphology analysis on neurons in mouse amygdala expressing Pgap2-targeting shRNA or overexpressing miR-483-5P (new Fig 3g-h, text page 5). The changes of the dendritic spine morphology observed *in vivo*, when compared to what we observed *in vitro*, differed slightly in the proportion of morphological spine subclasses, which was to be expected. Most importantly, however, our main conclusion that overexpression of miR-483-5p (and similarly downregulation of Pgap2) produces an increase in the proportion of mushroom spines holds regardless of the preparation used.

Question. Need to add an *in vivo* validation of the overexpression of miR-483-5P that is described in figure 6. It should also be shown *in vivo* that this leads to down regulation of Pgap2. Similarly, the shRNA to Pgap2 experiments are missing *in vivo* validation.

Answer. The requested experiments are now included in the new Supplementary figure 5 (text pages 5, 6, 7 and 9). We found that upon lentiviral-driven over expression of miR-483-5P (now verified *in vivo* in panel Supplementary Figure 5a), or *in vivo* expression of Pgap2-targeting shRNA, the level of Pgap2 mRNA is significantly reduced in mouse amygdala (panel Suppl. Fig 5b-c).

Question. The authors should include the full list of stress-regulated miRNAs in the supplemental material.

Answer. The list of all statistically significantly affected miRNAs is presented as a part of Figure 1E-F. There were only 5 affected miRs and they are all listed in this Figure.

Question. Figure 5 a, looks like a typo: is Macf1 3'UTR is 22,074 pb?

Answer. The typo has been corrected (now Figure 6).

Question. Why was a different amount of nucleotides (1-3) mutated in the different 3' UTR tested in figure 5?

Answer 5. To abolish the miR-target sequence interaction we performed a site-directed mutagenesis protocol to mutate the regions critical for miR-binding. In our hands the protocol generated various numbers of mutations within the miR-interacting sequences, having the same functional consequences - the abolishment of mir-target interaction.

Question. The authors should have used more than one behavioural test as a phenotypic readout of anxiety and stress.

Answer . We took advantage of the Elevated-Plus Maze paradigm as the one most commonly used for the measurement of anxiety-like behaviour. Preparation of animal cohorts to be used for other behavioural tests would require at least doubling the number of animals that underwent invasive surgical procedures and induction of restraint stress and would not advance our studies mechanistically. Rather than performing these difficult to ethically justify experiments, we have focused on extending the mechanistic insight of our studies by performing similar behavioural experiments using a cohort of animals in which Pgap2 had been knocked down (Fig 7g-h, 8d-e). We believe that all our EPM data is of sufficient quality, convincingly demonstrating the anxiolytic effects of miR-483-5p is driven by Pgap2 degradation (Figures 7 and 8).

Reviewer #2

Question. There is a lack of information regarding the results of the statistical analyses for all the data presented in the manuscript. Only p values are mentioned. The only exception is for the data shown in Figure 2a, where an F value with its corresponding degrees of freedom is provided. Without this information, it is not possible to determine the validity of the findings presented.

Answer. We agree with the fair criticism from the Reviewer. We now provide the description of the detailed statistical analyses in the related results parts of the text and in some cases in the figures' legends.

Question. An experiment demonstrating co-expression of miR-483-5p and Pgap2 in BLA neurons of adult mice is missing.

Answer. We have now included the requested experiment in the Figure 2a-b (text page 3). We demonstrated by in situ hybridisation, followed by immunohistochemistry, the co-localisation of the miR-483-5p with the neuronal marker NeuN, synaptic marker HOMER and PGAP2 in mouse amygdala.

Question. The main conclusion of the study is that “miR-483-5p offsets functional and behavioural effects of stress through synapse-targeted repression of Pgap2 in the basolateral amygdala”. However, this conclusion is not accurate because:
-first, the study does not provide (in vivo) evidence of (a) alterations in dendritic and/or spine structure in BLA neurons of mice subjected to restrain stress and (b) of attenuation of these structural changes by miR-483-5p over-expression in BLA neurons
-second, the study does not provide (in vivo) evidence that miR-483-5p overexpression in BLA neurons of mice downregulates Pgap2 expression

Answer. We demonstrated stress-induced decrease in the proportion of mushroom-like spines in the wild type mouse basolateral amygdala in our previous publication (Skrzypiec et al., 2013, Plos One). Here, we found that stress-induced upregulation of miR-483-5p counteracts these changes, by markedly increasing the proportion of mushroom-like spines, both in vitro and in vivo (Fig 3d-e and new Fig 3 g-h). Downregulation of the miR-483-5p target, Pgap2, recapitulates these observations. The conclusion of Pgap2-mediated, anti-stress effect of miR-483-5p is further corroborated by an anxiolytic effect of either overexpression of miR-483-5p or downregulation of Pgap2, with or without stress. Moreover, following the Reviewer’s request, we have now demonstrate that upon miR-483-5p or anti-Pgap2 shRNA the level of mRNA coding for Pgap2 is significantly downregulated (Supplementary Figure 5b-C, text pages 5, 6, 7 and 9).

Question. In vivo confirmation of overexpression of miR-483-5p and of downregulation of Pgap2 in BLA amygdala neurons in the lentiviral experiments is missing. These changes were only tested in vitro in a separate assay

Answer. We have now included the results of the requested experiments in the Supplementary Figure 5 (text page 6). They clearly confirm the validity of our constructs in vivo.

Question. The lentiviral manipulations were not designed to target amygdala neurons selectively.

Answer. We verified the precision of the lentiviral injections and spatial restriction of the viral expression by immunohistochemistry against the GFP co-expressed within the same construct (Figure 7b). We did not aim to restrict the expression further by using a neuronal subtype-specific promotor.

Specific points:

Statistical analysis:

Question. The sample size for the qPCR experiments (n=3) is low

Answer: In the novel requested experiments we increased the n numbers accordingly. In the case of animal-based studies we are bound by ethical regulations allowing us to use the minimal number of the animals to obtain statistically significant results as the part of the 3R policy.

Question. Data should be plotted as individual values

Answer: In the figures presenting the behavioural studies (Figures 7 and 8) we plotted each data point corresponding to every animal individually.

Question. Abstract: It is not clear if the authors are referring to humans and/or rodents

Answer: We now clarify this uncertainty

Question. The following statements are not supported by the results (as explained in the overall comments):

“Here we show that miR-483-5p in the amygdala counterbalances the structural, functional and behavioural consequences of psychological stress to promote anxiety” and

“Our results demonstrate that miR-483-5p is sufficient to confer an anxiolytic effect and point to miR-483-5p-mediated repression of Pgap2 as a critical cellular event

offsetting the functional and behavioural consequences of psychological stress”.

Answer: The issues raised in the point above have been responded in the “overall comments” above.

Question. Introduction: The references cited do not always correspond to the text preceding them (for instance see the first sentence)

Answer: The text preceding the reference has been modified (Page 2)

Question. The authors state that “The molecular functioning of central neurons is orchestrated by small, noncoding RNA sequences called microRNAs”. However, not only microRNAs orchestrate this function.

Answer: Thank you for the comment. We changed the text to emphasize that microRNAs are one of the modulators of the central nervous system function (Page 2)

Question. It is not clear what “pathway fine-tuning” refers to and how exactly miRNAs contribute to this.

Answer: We clarified our statement limiting it to the regulation of spatiotemporal gene expression levels (Page 2)

Question. The authors need to specify the species they are referring to (rodents and/or humans)

Answer: We clarified in the text that we refer to mouse studies (Page 3)

Question. It is mentioned that the molecular and cellular mechanisms by which microRNAs “regulate stress resilience are largely unknown”. It seems to me that the same is the case for stress susceptibility. The introduction does not provide a clear rationale for the study and does not mention the study’s goal.

Answer: We clarified our statement (Page 3) and provided the study’s goal

Question. Results: The sex, age, and strain of mice needs to be mention right from the start. This is only mentioned in the method sections, at the end of the manuscript

Answer: We provided the requested information (Page 3)

Question. Regarding the bioinformatics approach, more detail is needed about the criteria that was used to identify potential gene targets. For instance, it is not clear how exactly “stress-related” targets were identified.

Answer: AmiGo gene ontology software allows the selection of the biological pathways the candidate gene plays a role in. In order to identify the potential genes related to psychological stress, we filtered out the candidate genes considering their known and predicted role in stress.

Question. Are there genes that are not altered after dexamethasone exposure in the in vitro studies?

Answer: The list of genes we analysed is included in Supplementary Figure 1. The genes showing almost no change in level of expression upon dexamethasone treatment were Cul4a and Rbpj.

Question. Regarding the immunofluorescence image, the quality is low and critical information is missing, including antibody specificity and phenotype of labelled cells.

Answer: We replaced the low-quality images with better ones (Figures 2A-B, 6C), the epitope recognised by the antibodies allowing cell-type identification are included in figure legends and in “Immunohistochemistry” chapter of the Materials and Methods

Question. Fig 4: without showing the precise statistical analysis and the individual data, it is difficult to determine whether stress alters the expression of Gpx3 and Macf1 in the synaptosomal fraction

Answer: We provided a more detailed description of the statistical analysis we used in the aforementioned experiment (Page 4)

Reviewer #3

Major Concerns:

Question. Did the 5 mRNAs that are mentioned from the discovery microarray have a level of significance that survived correction for multiple testing? Is the $P < 0.001$ level the nominal significance level or the p-value after correction?

Answer: The significance values for the results of the microarray are presented as p-values adjusted by Bonferroni correction method (Page 3)

Question. After validation, the authors state, “Therefore, our subsequent studies focused on the role of amygdalar miR-483-5p in the stress response and on its mechanism of action.” But it is not clear why they focused on miR-483-5p over the 4 other candidates that were most differentially expressed, and appeared to all similarly survive validation.

Answer: Although, undoubtedly it would provide more in-depth understanding of the multi-miR effects on the stress response we decided to focus on the role of the miR-483-5P as we found this molecule to be the most upregulated in response to our anxiety inducing paradigm - acute, 6h restraint stress.

Question. Figure 2e – example figures of the dendritic change: the level of contrast / brightness / sharpness of the two figures looks very different. Specifically the miRNA transfected example looks blurry and a simple interpretation of the quantitative result is that if they were not equivalent images, the miRNA may appear to have more mushroom-like and fewer filopodia, but that may be because one can't visualize the filopodia with blurry pictures.

Answer: We have now replaced the example figures with ones of a higher quality (Figure 3e).

Question. Does the shRNA suppression of Pgap2 lead to the same cellular / dendritic / spine changes that overexpression of miR-483-5p does?

Answer: In addition to in vitro studies we have now performed in vivo experiments to verify our findings and respond to the reviewer's concern (new Fig 3g). Indeed, the pattern of changes seems to follow the same trend – the increase in percentage of mature mushroom spines at the expense of more plastic forms. Also, the dendritic spine density in both cases shows a significant increase when compared to controls (Figure 3g-h).

Question. Authors show that miR-483-5p is significantly upregulated following restraint stress but then go on to argue that it mediates a compensatory / resilient process via Pgap2, and that suppressing Pgap2, or overexpressing miR-483-5p, is

anxiolytic on a plus maze. However, the logic is not complete, in that they identified the miR following restraint stress, and generally stress is going to increase a number of anxiety / fear / stress behaviors, but they show that the miR is anxiolytic. To argue that it is a compensatory process, it would seem important to show that either inhibiting miR-483-5p (e.g. with a miR sponge) or overexpressing miR-483-5p would increase or decrease, respectively, restraint stress-dependent behaviors, based on the initial observation.

Answer: We agree with the fair criticism from the Reviewer and the appropriate, requested experiments have been included in Figure 8. We found that indeed, expression of miR-483-5P in mouse amygdala prevented angiogenesis. Our observation was additionally mechanistically expanded by expression of Pgap2-targeting shRNA in mouse amygdala, demonstrating the same behavioural phenotype.

Minor Concerns:

Question. It is not clear how long after the restraint stress that animals were sacrificed for miRNA analysis

Answer: Animals were sacrificed immediately at the end of the 6h-long session of restraint stress. We have clarified this in the Material and Methods section (Page 10).

Question. In the first section of results, the authors state, “We then observed that stress caused a further 4-fold increase of miR-483-5p in the synaptosomal fraction, but not in the cytosol (Figure 2a, $F(3, 11) = 23.11$, $P < 0.00005$, $p < 0.001$, $n = 3-4$ per group).” – it is not clear why two different p-values are presented here.

Answer: P values refer to both analyses performed before and after stress – synaptosomal and cytosolic fractions respectively. We corrected the description to clarify the statistical tests we performed (Page 4).

Question. There were several typographical errors throughout; I encourage a careful rereading / editing for typographical errors and grammar.

Answer: We accept this fair criticism and have done our best to improve grammatical and typographical errors.

Reviewers' comments:

Reviewer #1 (Remarks to the Author):

The authors did an excellent job addressing the reviewers concerns.

Reviewer #4 (Remarks to the Author):

The manuscript "miR-483-5p offsets functional and behavioural effects of stress through synapse targeted repression of Pgap2 in the basolateral amygdala" uses biochemistry, cell morphology, and mouse behavior to demonstrate exciting new roles for the microRNA miR-483-5p and its target gene Pgap2 in anxiety-related behavioral responses to stress. The manuscript contains robust and exciting findings that are sufficiently novel and of broad interest to the field. It is generally well written and, with a major exception described below, the data are correctly analyzed and interpreted. This is my first time reviewing the manuscript, but I can see that previous reviewers pointed out a number of concerns. The authors have done an excellent job addressing those concerns. Of particular note are the excellent new experiments showing *in vivo* changes in dendritic spine morphology (Figure 3) and validation of the effects of viral manipulations on Pgap2 expression (Supp Figure 5). I thus feel that the manuscript is experimentally complete, and I do not think that further experiments are required, again with one possible major exception described below.

Major Concerns

1) Although the manuscript provides a wealth of new and exciting data, these data are overinterpreted in the title and throughout the manuscript. The study shows that:

- a. miR-483-5p is upregulated in amygdala by stress
 - b. miR-483-5p represses Pgap2
 - c. miR-483-5p in amygdala increases mature spines and reduces anxiety-like behavior
 - d. Repression of Pgap2 in amygdala increases mature spines and reduces anxiety-like behavior
- However, taken together, these findings do not conclusively demonstrate that the effects of amygdalar miR-483-5p on spines and behavior occur THROUGH Pgap2 repression. In order to do this, the authors must overexpress miR-483-5p in amygdala along with overexpression of a Pgap2 construct that lacks the miR target seed and show that this prevents the effects of miR-483-5p on spines and behavior. Without such an experiment, it remains possible that miR-483-5p is actually working through other targets to drive changes in spines and behavior. This is a tall order experimentally, and I do not necessarily think this is required for the manuscript to be impactful for the field. However, without such an experiment, the interpretation of the data (in the title, the abstract, and throughout the manuscript) must be very much softened to suggest that amygdalar miR-483-5p COULD be working through Pgap2 in the amygdala, but future studies will be required to make that connection definitive.

Reviewer #5 (Remarks to the Author):

The authors provided ample additional analysis and data in response to reviewer comments. The authors have sufficiently addressed reviewer concerns.

Minor issue:

Within the manuscript, the 'anxiety index' is poorly justified. Please provide additional rationale and citations to justify the use of the anxiety index versus more traditional measurements such as time in open arm.

Many thanks to reviewers for their valuable comments. We performed additional experiments as requested by the reviewer #4 – our data confirm that effects of miR-483-5p in amygdala on spines and behavior indeed occur through Pgap2 repression. All the changes are highlighted in yellow throughout the manuscript – and new data can be found on Figure 3g and Figure 8.

REVIEWER COMMENTS

Reviewer #1 (Remarks to the Author):

The authors did an excellent job addressing the reviewers concerns.

Reviewer #4 (Remarks to the Author):

The manuscript “miR-483-5p offsets functional and behavioural effects of stress through synapse targeted repression of Pgap2 in the basolateral amygdala” uses biochemistry, cell morphology, and mouse behavior to demonstrate exciting new roles for the microRNA miR-483-5p and its target gene Pgap2 in anxiety-related behavioral responses to stress. The manuscript contains robust and exciting findings that are sufficiently novel and of broad interest to the field. It is generally well written and, with a major exception described below, the data are correctly analyzed and interpreted. This is my first time reviewing the manuscript, but I can see that previous reviewers pointed out a number of concerns. The authors have done an excellent job addressing those concerns. Of particular note are the excellent new experiments showing in vivo changes in dendritic spine morphology (Figure 3) and validation of the effects of viral manipulations on Pgap2 expression (Supp Figure 5). I thus

feel that the manuscript is experimentally complete, and I do not think that further experiments are required, again with one possible major exception described below.

Major Concerns

1) Although the manuscript provides a wealth of new and exciting data, these data are overinterpreted in the title and throughout the manuscript. The study shows that:

- a. miR-483-5p is upregulated in amygdala by stress
- b. miR-483-5p represses Pgap2
- c. miR-483-5p in amygdala increases mature spines and reduces anxiety-like behavior
- d. Repression of Pgap2 in amygdala increases mature spines and reduces anxiety-like behavior

However, taken together, these findings do not conclusively demonstrate that the effects of amygdalar miR-483-5p on spines and behavior occur THROUGH Pgap2 repression. In order to do this, the authors must overexpress miR-483-5p in amygdala along with overexpression of a Pgap2 construct that lacks the miR target seed and show that this prevents the effects of miR-483-5p on spines and behavior. Without such an experiment, it remains possible that miR-483-5p is actually working through other targets to drive changes in spines and behavior. This is a tall order experimentally, and I do not necessarily think this is required for the manuscript to be impactful for the field. However, without such an experiment, the interpretation of the data (in the title, the abstract, and throughout the manuscript) must be very much softened to suggest that amygdalar miR-483-5p COULD be working through Pgap2 in the amygdala, but future studies will be required to make that connection definitive.

Answer:

We'd like to thank the reviewer #4 for their valuable comment. As requested by the reviewer we performed additional experiments – we overexpressed miR-483-5p in amygdala along with *Pgap2* resistant to miR-483-5p and showed that miR-483-5p's effects on both spine morphology and behavior require suppression of PGAP2 (Figure 3g and Figure 8).

Reviewer #5 (Remarks to the Author):

The authors provided ample additional analysis and data in response to reviewer comments. The authors have sufficiently addressed reviewer concerns.

Minor issue:

Within the manuscript, the 'anxiety index' is poorly justified. Please provide additional rationale and citations to justify the use of the anxiety index versus more traditional measurements such as time in open arm.

Answer:

We included into the methods section how anxiety index is calculated. Anxiety index, calculated as the proportion of open arm to total arm entries, or % of open arm entries, are preferred measures of anxiety in the EPM since the time spent in the open arms can be obscured by freezing behavior evoked by restraint stress – both arm entries or time have been suggested to be a good measure of anxiety in the EPM (Walf and Frye, Nature Protocols 2007).

Reviewers' comments:

Reviewer #4 (Remarks to the Author):

The authors did an incredible job of addressing my main concern by using a miR-483-5p-resistant Pgap2 to demonstrate causality. I believe this manuscript is now very much appropriate for publication.